# OpenGPT-4o-Image: A Comprehensive Dataset for Advanced Image Generation and Editing

**Zhihong Chen** [* 1]  **Xuehai Bai** [*]  **Yang Shi** [* 2 3]  **Chaoyou Fu** [4]  **Huanyu Zhang** [5]  **Haotian Wang** [6]  **Xiaoyan Sun** [1 7]
**Zhang Zhang** [5]  **Liang Wang** [5]  **Yuanxing Zhang** [2]  **Pengfei Wan** [2]  **Yi-Fan Zhang** [5]

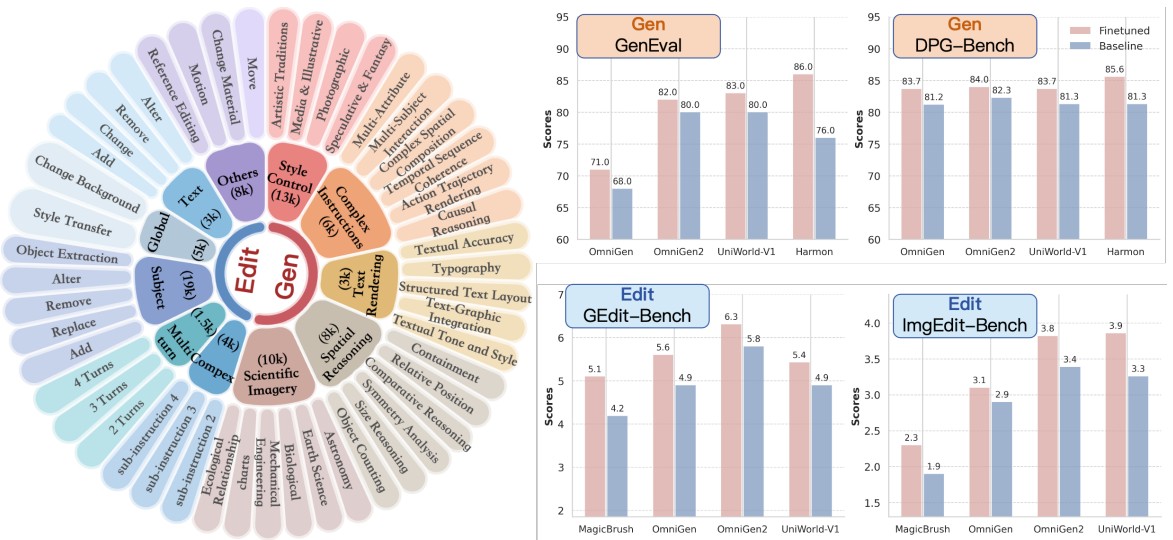

## Abstract

The performance of unified multimodal models for image generation and editing is fundamentally constrained by the quality and comprehensiveness of their training data. While existing datasets have covered basic tasks like style transfer and simple object manipulation, they often lack the systematic structure and challenging scenarios required for real-world applications. To address this bottleneck, we introduce **OpenGPT-4o-Image**, a large-scale dataset constructed using a novel methodology that combines hierarchical task taxonomy with automated data generation. Our taxonomy not only includes fundamental capabilities such as text rendering and style control but also introduces highly practical yet challenging categories like **scientific imagery** for physics/chemistry illustrations and **complex instruction editing** requiring simultaneous execution of multiple operations. Through an automated pipeline leveraging structured resource pools and GPT-4o, we generate 80k high-quality instruction-image pairs with controlled diversity, covering 11 major domains and 51 subtasks. Extensive experiments show that fine-tuning leading models on our dataset achieves significant performance gains across multiple benchmarks, with improvements of up to 18% on editing tasks (UniWorld-V1 on ImgEdit-Bench) and 13% on generation tasks (Harmon on GenEval). Our work demonstrates that systematic data construction is key to advancing multimodal AI capabilities.

[*]Equal contribution  [1]MoE Key Lab of BIPC, NEL-BITA, University of Science and Technology of China [2]Kling Team, Kuaishou Technology [3]Peking University [4]Nanjing University [5]The Institute of Automation, Chinese Academy of Sciences [6]Tsinghua University [7]APKL of BIIP, IAI,Hefei Comprehensive National Science Center. Correspondence to: Yuanxing Zhang <longo11070001@gmail.com>, Yi-Fan Zhang <yifanzhang.cs@gmail.com>.

*Proceedings of the $43^{rd}$ International Conference on Machine Learning*, Seoul, South Korea. PMLR 306, 2026. Copyright 2026 by the author(s).

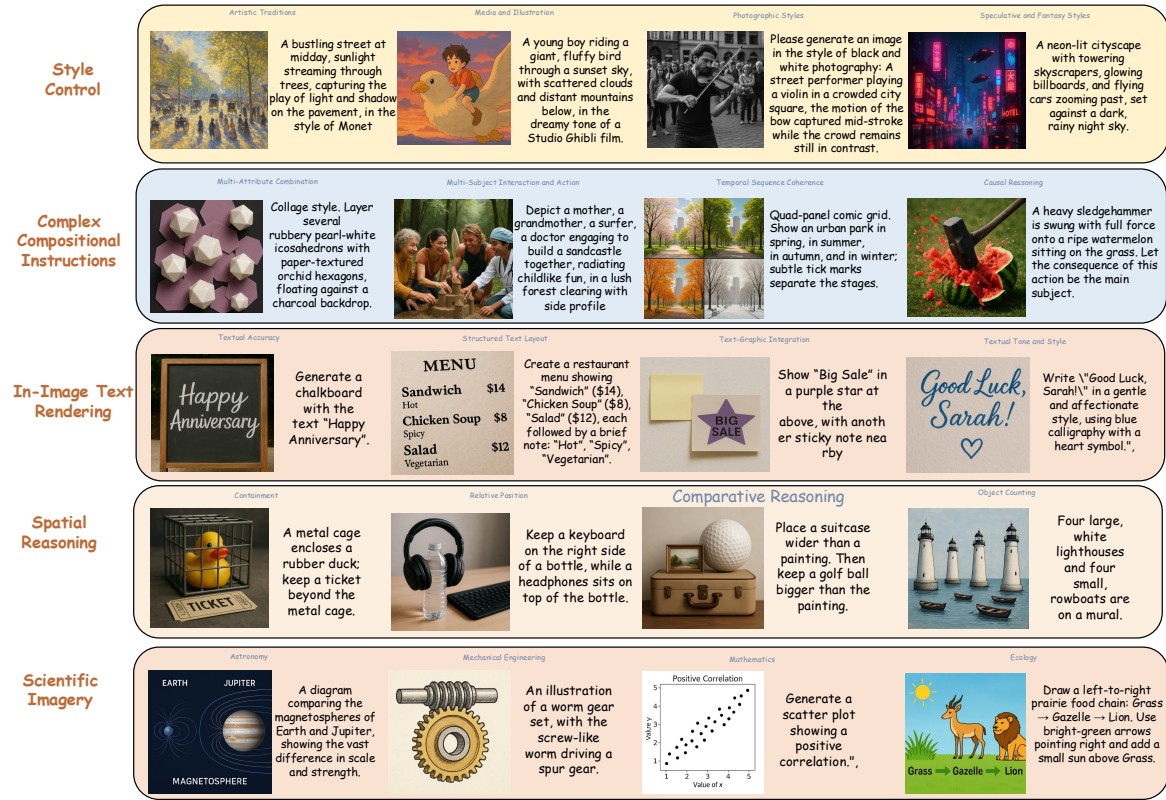

*Figure 1.* **Illustrative examples of image generation from OpenGPT-4o-Image.** We comprehensively categorize image generation tasks into five groups based on the core capabilities they target: (a) Style Control, focusing on rendering diverse artistic and aesthetic styles; (b) Complex Instruction Following, which tests the model's ability to adhere to intricate compositional and logical constraints; (c) In-Image Text Rendering, involving the accurate generation and placement of text within images; (d) Spatial Reasoning, which demands geometric precision in tasks like object counting and relative positioning; and (e) Scientific Imagery, extending the model's application to specialized domains such as science, engineering, and data visualization.

## 1. Introduction

The field of AI-powered content creation is being transformed by unified multimodal models capable of both generating and editing images from natural language instructions (Deng et al., 2025; Lin et al., 2025; Liu et al., 2025; Ma et al., 2026; Chen et al., 2026),. Despite these achievements, creating training data that comprehensively addresses the full spectrum of real-world applications remains challenging. Existing datasets have made valuable contributions in areas such as style transfer, basic object manipulation, and increasingly, **text rendering** (Wu et al., 2025b)—a capability that has rightfully gained attention for its practical importance. However, certain complex scenarios that require specialized knowledge or sophisticated reasoning still present difficulties for current models. For instance, tasks involving technical illustrations for scientific education, or editing operations that require executing multiple instructions simultaneously, often reveal limitations in model capabilities. These challenges suggest opportunities for more structured approaches to data collection and taxonomy design.

To help address these opportunities, we introduce **OpenGPT-4o-Image**, a dataset designed to support the development of more capable and robust multimodal systems. Our work builds upon previous research while expanding coverage to include additional challenging scenarios. The dataset is organized around a hierarchical taxonomy that systematically addresses both established challenges and less explored areas. As shown in Figure 1 and 4, this includes not only refining existing focus areas like **style control** and **text rendering**, but also introducing categories such as: *1. Scientific Imagery*: Supporting technical illustration needs in fields like physics, chemistry, and biology, where visualizations play a critical role in education. *2. Complex Instruction Editing*: Addressing scenarios where users naturally provide multiple editing instructions that should be executed in concert. *3. Spatial Reasoning and Causal Inference*: Expanding beyond basic object recognition to include more sophisticated relational understanding.

To ensure scalability and consistency, we develop an automated pipeline that generates high-quality instruction-image pairs. This approach allows us to create 80,000 samples spanning 11 major domains and 51 subtasks with controlled

diversity and difficulty levels. Our experimental results demonstrate the utility of this approach. When fine-tuned on OpenGPT-4o-Image, several leading models show consistent improvements across multiple benchmarks. For example, UniWorld-V1 (Lin et al., 2025) achieves a 18% relative improvement on ImgEdit-Bench (Ye et al., 2025b), while Harmon (Wu et al., 2025d) shows a 13% gain on GenEval (Ghosh et al., 2024). These improvements suggest that our structured approach to data construction can help models better handle complex and specialized tasks. In summary, our primary contributions are:

◇ A hierarchical taxonomy for image generation and editing that systematically decomposes complex tasks into 51 fine-grained sub-capabilities across 11 major domains. For image generation, this includes five core modules: *Style Control*, *Complex Instruction Following*, *In-Image Text Rendering*, *Spatial Reasoning*, and *Scientific Imagery*. For editing, we define six categories with 21 subtasks, including *Subject Manipulation*, *Text Editing*, *Complex Instruction Editing*, *Multi-Turn Editing*, *Global Editing*, and other challenging forms.

◇ An automated, scalable pipeline for generating high-quality training data that GPT-4o to produce 80k instruction-image pairs with controlled diversity and difficulty levels. Our pipeline ensures comprehensive coverage of both fundamental capabilities and challenging scenarios through systematic task definition and template-based generation.

◇ Our experimental results demonstrate the substantial utility of our dataset and methodology. We employ four leading models spanning different architectural paradigms—including UniWorld-V1, Harmon, Omni-Gen2, and MagicBrush—to ensure the generalizability of our findings. The models are evaluated on four standardized benchmarks: GEdit-Bench and ImgEdit-Bench for image editing capabilities, and GenEval and DPG-Bench for text-to-image generation quality. The evaluation reveals consistent and significant improvements across all tested configurations.

We hope that OpenGPT-4o-Image will contribute to future research by providing a resource that addresses a broader range of real-world applications, and that our methodology inspires further work on systematic data construction for multimodal AI.

## 2. Method

OpenGPT-4o-Image encompasses a broader spectrum of categories, more precise and comprehensive instructions, and a substantial collection of practical as well as challenging generation and editing tasks. Most importantly, the quality

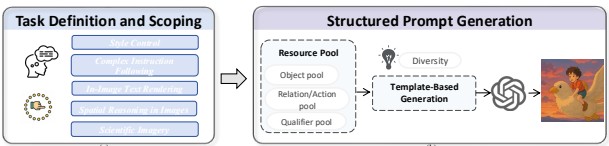

*Figure 2.* **Image Generation Data Construction Pipeline.** (a) illustrates the Task Definition and Scoping phase, where target capabilities are precisely defined, decomposed into hierarchical categories, and assigned difficulty grades. (b) shows the Structured Prompt Generation phase, where instructions are generated at scale by populating diverse syntactic templates with components from structured resource pools, which are used to produce the image.

achieved in both image generation and editing is remarkable. Section 2.1 and 2.2 enumerate the categories of image generation and editing, while Section 2.3 and 2.4 provide a detailed account of the data pipeline. Additional details on the dataset distribution are provided in the Appendix B.

### 2.1. Generation Type Definition

**Style Control.** This module is designed to enhance the model's ability to render a diverse range of visual styles. Comprising a substantial 13k samples, it is organized into a comprehensive collection of four distinct categories. The first, Artistic Traditions, covers historical and cultural art forms, including Western movements (Impressionism, Post-Impressionism, Cubism), Eastern traditions (Japanese Ukiyo-e, Ink Wash Painting, Traditional Chinese Painting), and contemporary forms like Graffiti Art. The second category, Media and Illustration, focuses on aesthetics from popular media and illustration, spanning animation (Ghibli Animation Style, Pixar Animation Style, Classic Animation Style), comics (Japanese Manga Style), digital-native styles (Pixel Art, Block-Based Art, Mosaic Style), and foundational techniques like Sketch and Line Art. The third, Photographic Styles, is dedicated to the visual language of photography, encompassing technical styles (Analog Film Aesthetic, High Dynamic Range (HDR)), artistic choices (Monochrome Photography), and mood-based aesthetics (Ethereal and Dreamlike, Vintage and Retro Aesthetics). Finally, Speculative and Fantasy Styles explores fictional genres such as Cyberpunk, Steampunk, Cosmic and Space Opera, and Digital Futurism.

**Complex Instruction Following.** This module, comprising a focused set of 6k samples, addresses the model's capacity for precise instruction adherence, particularly for prompts involving multiple constraints and complex logical relationships. We decompose this capability into several specific sub-tasks. These include the composition of static scenes through Multi-Attribute Combination and Multi-Subject Interaction and Action, as well as arranging elements according to a Complex Spatial Composition. Furthermore, the module assesses the understanding of dynamic and sequential concepts via Temporal Sequence Coherence and

Action Trajectory Rendering. Finally, it probes the model's Causal Reasoning abilities.

**In-Image Text Rendering.** This module addresses in-image text rendering, a capability of high practical value that has been largely unaddressed in prior instruction-following datasets. It is designed around a core of 3k samples to systematically improve the model's ability to create accurate and aesthetic text-graphic compositions. The module covers foundational aspects such as Textual Accuracy (verbatim content rendering) and Typography (font control). It then progresses to more complex structural and relational challenges, including Structured Text Layout for multi-line arrangements and Text-Graphic Integration for the coherent placement of text within the image. Finally, the scope is expanded to include Multilingual Support for non-English scripts and the nuanced task of aligning Textual Tone and Style with the image's overall aesthetic.

**Spatial Reasoning.** In contrast to the first three modules, which primarily emphasize semantic and aesthetic interpretation, the Spatial Reasoning component focuses on the model's understanding of fundamental spatial and logical relationships. Consequently, tasks in this section demand geometric precision and logical fidelity. To assess this capability, we utilize a dedicated dataset of 8k samples divided into several categories, including basic topological relations (Containment) and discrete 2D Relative Position. The model's numerical capacity is tested via Object Counting, while its understanding of geometric properties and comparisons is evaluated through Size Reasoning, Symmetry Analysis, and Comparative Reasoning.

**Scientific Imagery.** The Scientific Imagery module extends the applicability of text-to-image models to specialized professional domains, addressing the growing demand for image generation in scientific research and education where relevant training data remains scarce. To fill this gap, we provide 10k samples covering a breadth of disciplines, including Mathematics, Physics, and Mechanical Engineering; natural sciences such as Astronomy and Earth Science; life sciences like Biological studies and Ecology; and topics from Culture and History. The introduction of this module provides a data foundation for exploring the model's potential in professional knowledge visualization and its applications in scientific communication and education.

### 2.2. Edit Type Definition

We define six categories of editing tasks: Subject Manipulation, Text Editing, Complex Editing, Multi-Turn Editing, Global Editing, and other challenging forms of editing. More specifically, these categories are further divided into 21 subtasks, which collectively encompass a wide range of practical instruction-based image editing scenarios.

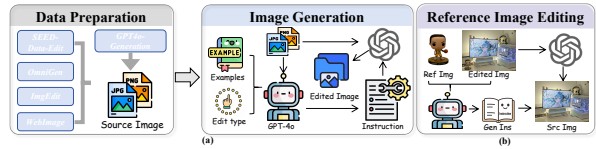

*Figure 3.* **Image Editing Data Construction Pipeline.** (a) illustrates the source image acquisition and demonstrates the use of GPT-4o for generating instructions and producing the edited image. (b) shows the process of reference image editing, where Subject-Driven Image Generation is used to create reference and edited images, with GPT-4o applied for inpainting the source image.

**Subject Manipulation.** Building on classical instruction-based image editing, we define we introduce Subject Manipulation, which focuses on precise local modifications of specified target objects while preserving the integrity and consistency of the background and other semantically irrelevant regions. Subject Manipulation encompasses five operations: Add, Remove, Replace, Alter, and Object Extraction and comprises 19k samples. Specifically, Alter refers to modifications that change only the attributes of an object, whereas Replace involves a comprehensive transformation of the object itself. Additionally, Add and Remove denote the insertion or deletion of an object, respectively, whereas Object Extraction refers to isolating and extracting a specific object from the image.

**Text Editing.** Given the substantial gap between GPT-4o (Hurst et al., 2024) and existing open-source unified models in terms of text editing capabilities, we define Text Editing as the task of modifying text elements embedded within images. In contrast to conventional image editing, which often focuses on coarse-grained objects, text editing requires a deeper comprehension of the rich semantics embedded in editing instructions and finer-grained editing capabilities. Drawing inspiration from Subject Manipulation, we further define four types: Text Add, Text Replace, Text Alter, and Text Remove, including 3k samples.

**Complex Instruction Editing.** With the increasing demand for image editing, users often issue multiple instructions that they expect to be executed simultaneously. However, under such scenarios, existing models frequently suffer from limited instruction-following capabilities and reduce image generation quality. To address this limitation, we introduce the task of Complex Instruction Image Editing, comprising a total of 4k samples, designed to enhance the ability of unified models to handle complex instructions. Furthermore, we categorize complex instructions based on their level of complexity. Each complex instruction consists of two to four distinct sub-editing operations, which are drawn from Text Editing and Subject Manipulation except Object Extraction.

**Multi-turn Editing.** Building on the insights from interactive text modifications in multimodal large language models, we extend the concept of multi-turn interactions to the

domain of image editing. In this context, we introduce Multi-Turn Image Editing, including 1,500 samples, where iterative user interactions progressively guide the modification of image content. Specifically, we categorize the task by the number of interaction rounds, distinguishing between two-round, three-round, and four-round editing scenarios. This task is expected to strengthen unified models in multi-turn image editing and to achieve controllable editing aligned with user satisfaction.

**Global Editing.** In addition to local image editing, we also introduce global editing, comprising a total of 4k samples, which encompasses Background Replacement and Style Transfer. For style transfer, we meticulously design 11 distinct styles, including Cyberpunk Style, Ghibli Style, Ink Painting Style, Disney Animation Style, Hand-drawn Style, Monai Style, and others. In contrast, background replacement operations focus on substituting the surrounding environment while maintaining the integrity of foreground objects. This editing type maintains the subject elements and their spatial relationships, ensuring seamless integration into entirely different backgrounds.

**Other Challenging Editing.** Beyond the aforementioned categories, we define a set of challenging editing tasks, including reference image editing, motion modification, material transformation, and object movement. Reference image editing involves the seamless incorporation of specified subjects into the original image. Motion modification emphasizes altering and adjusting the expressions and movements of objects, thereby enabling flexible control over their dynamic characteristics. Material transformation focuses on modifying the texture or composition of clothing and other materials. Finally, we introduce a novel task, Object movement, which entails repositioning an object from one location to another within the image while preserving spatial coherence and visual realism.

### 2.3. Automatic Dataset Pipeline for Image Generation

To systematically construct the large-scale, hierarchical training dataset for OpenGPT-4o-gen, we formulated and adhered to a general data construction pipeline, illustrated in Figure 2. This pipeline is designed to provide clear training objectives for each submodule while ensuring controlled diversity, a balanced difficulty distribution, and reliable data quality. It converts abstract capability targets into concrete, trainable data through two main stages.

**Task Definition and Scoping.** This initial phase marks the starting point of the construction process, where we clearly delineate the target capability for each submodule. We begin with Capability Definition and Boundary Setting, precisely defining the core skill each module aims to train. To ensure the purity of the training signal, we establish strict boundaries, specifying its scope and exclusions. For example,

in the "Relative Position" module, we focused on planar relations like left of and above, while intentionally excluding other spatial concepts like 3D perspective or topological containment. For more complex capabilities, we perform Hierarchical Categorization, decomposing them into logically interconnected sub-categories. In the "Causal Reasoning" module, for instance, we divided the task into levels such as "Explicit Causality," "Implicit Inference," and "Complex Causal Chains" to enable more granular training and evaluation. Finally, we implement Difficulty Grading for each submodule, assessing difficulty based on factors like instruction complexity, required background knowledge, and the length of the reasoning chain. This allows us to generate instruction sets with a well-distributed range of difficulty.

**Structured Prompt Generation.** Once the task definitions are established, this phase focuses on the efficient and high-quality generation of instruction text. We first implement Resource Pool Design, constructing a series of structured resource pools for each module to serve as foundational components for prompt generation. These pools typically include an Object Pool (various entities), a Relation/Action Pool (core verbs, prepositions, and their synonyms), and a Qualifier Pool (scenes, materials, adjectives) to increase the naturalness of the prompts. We then employ Template-Based Generation, designing multiple templates with diverse syntactic structures and populating them with randomly sampled components from the resource pools. This method allows us to generate instructions at scale that possess both structural consistency and linguistic variety. To further ensure data quality, we apply a Diversity Strategy, which includes varying the presentation format of instructions, controlling the combinatorial logic of content elements, and judiciously injecting content-aligned stylistic elements into a subset of prompts to enhance their realism.

### 2.4. Automatic Dataset Pipeline for Image Editing

**Data Preparation.** We construct the dataset corpus by integrating multiple high-quality sources, including SEED-Data-Edit (Ge et al., 2024), ImgEdit (Ye et al., 2025b), GPT-4o (Hurst et al., 2024) generated images, OmniEdit (Wei et al., 2024), and a curated collection of high-resolution images as the initial source corpus. Compared with existing datasets, our corpus contains source images of substantially higher quality, a broader spectrum of editing categories, more diverse instruction types, and correspondingly higher-quality edited results. To ensure dataset quality, we select distinct source corpora for each editing type. For text editing, GPT-4o is used to generate images with embedded textual elements, serving as the source corpus for text-related tasks. Reference image editing utilizes reference-target image pairs from the subject-driven generation component of OmniEdit, which are adopted as target and reference images in our corpus. Motion and material modifications are based

*Table 1.* Comparison of fine-tuning results of different models on our dataset on GenEval (Ghosh et al., 2024) benchmark. Results of GPT-4o are tested by (Yan et al., 2025).

| Method | Single object | Two object | Counting | Colors | Position | Color attribution | Overall |
|---|---|---|---|---|---|---|---|
| *Open-source Models* | | | | | | | |
| SDv2.1 (Rombach et al., 2022) | 0.98 | 0.5 | 0.44 | 0.85 | 0.07 | 0.17 | 0.50 |
| SDXL (Podell et al., 2023) | 0.98 | 0.74 | 0.39 | 0.85 | 0.15 | 0.23 | 0.55 |
| IF-XL | 0.97 | 0.74 | 0.66 | 0.81 | 0.13 | 0.35 | 0.61 |
| LUMINA-Next (Zhuo et al., 2024) | 0.92 | 0.46 | 0.48 | 0.70 | 0.09 | 0.13 | 0.46 |
| SD3-medium (AI, 2024) | 0.99 | 0.94 | 0.72 | 0.89 | 0.33 | 0.60 | 0.74 |
| FLUX.1-dev (Labs, 2024) | 0.99 | 0.81 | 0.79 | 0.74 | 0.20 | 0.47 | 0.67 |
| OmniGen (Xiao et al., 2025) | 0.98 | 0.84 | 0.66 | 0.74 | 0.40 | 0.43 | 0.68 |
| TokenFlow-XL (Qu et al., 2025) | 0.95 | 0.60 | 0.41 | 0.81 | 0.16 | 0.24 | 0.55 |
| Janus (Wu et al., 2025a) | 0.97 | 0.68 | 0.30 | 0.84 | 0.46 | 0.42 | 0.61 |
| Janus Pro (Chen et al., 2025d) | 0.99 | 0.89 | 0.59 | 0.90 | 0.79 | 0.66 | 0.80 |
| Emu3-Gen (Wang et al., 2024) | 0.98 | 0.71 | 0.34 | 0.81 | 0.17 | 0.21 | 0.54 |
| Show-o (Xie et al., 2024) | 0.98 | 0.80 | 0.66 | 0.84 | 0.31 | 0.50 | 0.68 |
| MetaQuery-XL (Pan et al., 2025) | - | - | - | - | - | - | 0.80 |
| BLIP3-o 8B (Chen et al., 2025b) | - | - | - | - | - | - | 0.84 |
| BAGEL (Deng et al., 2025) | 0.99 | 0.94 | 0.81 | 0.88 | 0.64 | 0.63 | 0.82 |
| OmniGen (Xiao et al., 2025) | 0.98 | 0.84 | 0.66 | 0.74 | 0.40 | 0.43 | 0.68 |
| OmniGen2 (Wu et al., 2025c) | 0.99 | 0.92 | 0.75 | 0.87 | 0.60 | 0.69 | 0.80 |
| UniWorld-V1 (Lin et al., 2025) | 0.99 | 0.93 | 0.79 | 0.89 | 0.49 | 0.70 | 0.80 |
| Harmon (Wu et al., 2025d) | 0.99 | 0.86 | 0.66 | 0.85 | 0.74 | 0.48 | 0.76 |
| *Proprietary Models* | | | | | | | |
| GPT-4o | 0.99 | 0.92 | **0.85** | **0.92** | 0.75 | 0.61 | 0.84 |
| *Finetuning* | | | | | | | |
| OmniGen | 0.99 | 0.88 | 0.68 | 0.79 | 0.45 | 0.44 | 0.71 |
| OmniGen2 | 0.99 | 0.93 | 0.76 | 0.88 | 0.66 | 0.72 | 0.82 |
| UniWorld-V1 | 0.99 | 0.96 | 0.82 | 0.88 | 0.60 | **0.73** | 0.83 |
| Harmon | **0.99** | **0.96** | 0.79 | 0.88 | **0.85** | 0.69 | **0.86** |

on original images from ImgEdit. For object removal tasks, we include a subset of carefully curated high-resolution images, each with a resolution exceeding 1024 pixels. Finally, for multi-turn and other editing tasks, we leverage the multi-turn corpus from SEED-Data-Edit.

**Instruction Generation.** We provide the original image, the editing type, and a set of in-context examples to facilitate prompt generation. For style transfer, we define 10 distinct styles and generate the associated instructions. Additionally, we provide sub-reference editing types for complex instruction-based images, including Subject Manipulation and Text Editing, and supply carefully designed examples to guide the generation of diverse instructions. For reference image editing, we draw inspiration from subject-driven image generation. As illustrated in Figure 3(b), we provide the reference image, the result generated from subject-driven generation, and several in-context examples, which are then used with GPT-4o to generate instructions. For multi-turn image editing, we define 10 subcategories of tasks, including Text Editing, Global Editing, and Subject Manipulation. We further provide in-context examples and generate interactive multi-turn editing instructions.

**Image Generation.** To construct the OpenGPT-4o-Image dataset, the core of our data generation process relied on the gpt-image-1 API, which was employed to regenerate or aug-

ment all image–instruction pairs. Furthermore, as illustrated in Figure 3(b), for subject-driven image editing, we propose a novel dataset construction strategy that utilizes inpainting instructions together with edited outputs to generate the corresponding source image. Finally, multi-turn image editing generates the results of each round progressively, ensuring the overall quality of the editing process.

## 3. Experimental Analysis

In this section, we conduct a comprehensive evaluation of our dataset. Section 3.1 provides a detailed specification of the baseline models, the evaluation benchmarks, and the experimental setup. Section 3.2 presents the results of data scaling experiments, where subsets of different sizes were sampled to assess the impact of scaling on performance trends. Section 3.3 provides both qualitative and quantitative analyses of the experimental outcomes. Section 3.4 further conducts a user study to analyze human preferences. Finally, Section 3.5 performs comprehensive ablation studies on different subsets of the generation and editing datasets.

### 3.1. Experimental Setting

**Baselines.** For the data scaling experiments, we employ UniWorld-V1 (Lin et al., 2025), which integrates Qwen2.5-VL (Bai et al., 2025) as the comprehension model and

FLUX-dev (Labs et al., 2025) as the vision generation model. For the comprehensive evaluations, we conducted extensive comparisons across different modeling paradigms, including diffusion-based and autoregressive frameworks. The evaluated models comprise UniWorld-V1, Harmon (Wu et al., 2025d), OmniGen2 (Wu et al., 2025c), MagicBrush (Zhang et al., 2023), and OmniGen (Xiao et al., 2025).

**Benchmarks.** Our evaluation methodology encompasses a thorough quality assessment for both image generation and editing datasets. For the image generation dataset, we comprehensively evaluate its quality using GenEval and DPG-Bench; GenEval specifically assesses compositionality, while DPG-Bench focuses on semantic alignment. Similarly, our image editing dataset is comprehensively evaluated across two widely used benchmarks: GEdit-Bench and ImgEdit-Bench. GEdit-Bench covers 11 distinct evaluation dimensions and assesses visual text and portrait editing, whereas ImgEdit-Bench further provides fine-grained evaluation of complex instruction-based image editing.

## 3.2. Data Scaling Experiments

To validate the effectiveness of our dataset, as shown in Table 7, we perform uniform sampling on subsets of the dataset at 20K, 30K, and 40K sizes. Since relying on a single benchmark can introduce bias and lead to incomplete evaluations, we employ two benchmarks for a comprehensive assessment. We also compute the overall average across both benchmarks and observe a consistent upward trend in the average performance as the dataset size increases. Additionally, given the relatively small increase in performance between 30K and 40K, we select the 40K dataset size.

## 3.3. Compressive Evaluations

**Performance Improvement on Image Editing Model.** As shown in Table 2 and Table 6, we finetune on MagicBrush (Zhang et al., 2023), OmniGen (Xiao et al., 2025), UniWorld-V1 (Lin et al., 2025), and OmniGen2 (Wu et al., 2025c). MagicBrush achieves improvements of 21.1% and 21.7% on ImgEdit-Bench and GEdit-Bench, respectively. OmniGen gains of 6.9% and 14.0% on the two benchmarks. UniWorld-V1 improves by 18.4% on ImgEdit-Bench and 12.0% on GEdit-Bench, while OmniGen2 achieved 12.7% and 8.8% improvements, respectively. The results indicate that leveraging more advanced comprehension and editing models, together with fine-tuning on our high-quality dataset, enables UniWorld-V1 and OmniGen2 to achieve notable improvements across multiple dimensions, such as Adjust, Add, Action, and Compose.

**Performance Improvement on Image Generation Model.** We fine-tune four representative models: OmniGen, OmniGen2, UniWorld-V1, and Harmon. As detailed in Table 1 and Table 8, all models consistently exhibit significant per-

formance improvements after this process. For instance, Harmon's performance surges by 13.2% on Geneval and 5.3% on DPG-Bench. Even Omnigen2, which shows the most modest gains, still achieves robust improvements of 2.5% and 1.9%. We hypothesize that Harmon's markedly superior performance compared to the other models stems from its efficient architecture and smaller 1.5B parameter size. Notably, these substantial results are achieved using our curated dataset of only 40k samples. This strongly demonstrates that our dataset's clear taxonomy and high-quality instructions enhance a model's text-to-image capabilities with remarkably high data efficiency, particularly in its precision for following complex instructions.

**Performance Improvement on Unified Dataset.** As shown in Table 9, we finetune UniWorld-V1 using both ShareGPT-4o and our image editing dataset. Our dataset achieves significant improvements, surpassing ShareGPT-4o by 3.2% on ImgEdit-Bench, 1.7% on GEdit-Bench, 1.2% on Geneval and 1.1% on DPG-Bench. Under the same training configuration and with less data, our dataset achieves superior performance in both quantitative and qualitative evaluations, owing to its more comprehensive categorization and richer set of editing instructions.

**Quantitative Comparison.** Due to the limitations of quantitative metrics in evaluating editing tasks, we further conduct qualitative evaluations to assess the effectiveness of our dataset, as illustrated in Figure. 6. Before finetuning, UniWorld-V1 exhibits suboptimal performance in following complex instructions, particularly in tasks involving object replacement and action modification. After efficient fine-tuning with the OpenGPT-4o-Image dataset, the model demonstrates substantial qualitative improvements. As shown in Figure 6, the fine-tuned UniWorld-V1 successfully replaces the hat with a teapot and raises the person's right arm, whereas the original model fails to execute these editing instructions effectively. Next, we turn to the Harmon model's improvements on text-to-image generation tasks. Here, the fine-tuned model demonstrates comprehensive performance gains, particularly in its ability to follow complex instructions. Specifically, we observe significant enhancements in handling various fine-grained capabilities, including in-image text rendering, multi-entity composition, temporal and spatial reasoning. These improvements collectively indicate that our dataset effectively strengthens the model's integrated understanding of multi-dimensional, complex semantics and its generation fidelity.

We conduct comprehensive quantitative and data scaling experiments, as detailed in Appendix C. These experiments demonstrate that fine-tuning with our image generation and editing dataset yields consistent improvements across a wide range of benchmarks, while also exhibiting qualitative superiority. Notably, our approach achieves substantial improve-

*Table 2.* Comparison of fine-tuning results of different models on our dataset on ImgEdit-Bench (Ye et al., 2025b). ‡ indicates results from our own tests without fine-tuning.

| Model | Add | Adjust | Extract | Replace | Remove | Background | Style | Hybrid | Action | Overall |
|---|---|---|---|---|---|---|---|---|---|---|
| *Open-source Models* | | | | | | | | | | |
| IP2P (Brooks et al., 2023) | 2.45 | 1.83 | 1.44 | 2.01 | 1.50 | 1.44 | 3.55 | 1.20 | 1.46 | 1.88 |
| AnyEdit (Jiang et al., 2025b) | 3.18 | 2.95 | 1.88 | 2.47 | 2.23 | 2.23 | 2.85 | 1.56 | 2.65 | 2.45 |
| UltraEdit (Zhao et al., 2024) | 3.44 | 2.81 | 2.13 | 2.96 | 1.45 | 2.86 | 3.76 | 1.91 | 2.98 | 2.70 |
| OmniGen (Xiao et al., 2025) | 3.47 | 3.04 | 1.71 | 2.94 | 2.43 | 3.21 | 4.19 | 2.24 | 3.38 | 2.96 |
| Step1X-Edit (Liu et al., 2025) | 3.88 | 3.14 | 1.76 | 3.40 | 2.41 | 3.16 | 4.63 | 2.64 | 2.52 | 3.06 |
| ICEdit (Zhang et al., 2025) | 3.58 | 3.39 | 1.73 | 3.15 | 2.93 | 3.08 | 3.84 | 2.04 | 3.68 | 3.05 |
| BAGEL (de Jong et al., 2006) | 3.56 | 3.31 | 1.70 | 3.30 | 2.62 | 3.24 | 4.49 | 2.38 | 4.17 | 3.20 |
| OmniGen2 (Wu et al., 2025c) | 3.57 | 3.06 | 1.77 | 3.74 | 3.20 | 3.57 | 4.81 | 2.52 | **4.68** | 3.44 |
| Ovis-U1 (Wang et al., 2025a) | 4.13 | 3.62 | **2.98** | **4.45** | **4.06** | **4.22** | 4.69 | **3.45** | 4.61 | **4.00** |
| FluxKontext dev (Labs et al., 2025) | 3.76 | 3.45 | 2.15 | 3.98 | 2.94 | 3.78 | 4.38 | 2.96 | 4.26 | 3.52 |
| MagicBrush (Zhang et al., 2023) | 2.84 | 1.58 | 1.51 | 1.97 | 1.58 | 1.75 | 2.38 | 1.62 | 1.22 | 1.90 |
| UniWorld-V1 (Lin et al., 2025) | 3.82 | 3.64 | 2.27 | 3.47 | 3.24 | 2.99 | 4.21 | 2.96 | 2.74 | 3.26 |
| OmniGen‡ (Xiao et al., 2025) | 3.27 | 3.05 | 1.90 | 2.82 | 2.43 | 3.00 | 4.11 | 1.67 | 3.27 | 2.90 |
| OmniGen2‡ (Wu et al., 2025c) | 3.60 | 3.44 | 1.94 | 3.81 | 2.54 | 3.79 | 4.57 | 2.71 | 4.49 | 3.39 |
| *Proprietary Models* | | | | | | | | | | |
| GPT-4o | 4.61 | 4.33 | 2.9 | 4.35 | 3.66 | 4.57 | 4.93 | 3.96 | 4.89 | 4.20 |
| *Finetuning* | | | | | | | | | | |
| MagicBrush | 2.66 | 2.12 | 1.45 | 2.04 | 2.03 | 2.18 | 2.04 | 2.19 | 1.60 | 2.30 |
| OmniGen | 3.75 | 3.48 | 2.17 | 2.68 | 2.03 | 3.23 | 4.17 | 2.87 | 3.81 | 3.10 |
| OmniGen2 | 4.15 | 3.69 | 2.53 | 4.11 | 3.64 | 4.10 | **4.69** | 2.99 | **4.68** | 3.82 |
| UniWorld-V1 | **4.34** | **4.28** | 2.66 | 3.92 | 3.30 | 4.15 | 4.62 | 3.43 | 3.97 | 3.86 |

### 3.4. Human Preference Evaluation

To verify that improvements on automatic metrics correspond to perceived quality, we conducted a human preference study. In this blind study, 20 expert evaluators were asked to select the best result from the outputs of four models on 400 samples: FluxKontext dev, UniWorld-V1 (our base model), ShareGPT-4o-Image (Janus-4o), and our own finetuned UniWorld-V1. The evaluators were not aware of the model identities. As shown in Table 3, our fine-tuned model was significantly preferred, demonstrating that our dataset imparts tangible gains in human-perceived edit faithfulness and visual quality. This also serves as an external validation against model bias, as our model outperforms strong proprietary baselines in a human-centric assessment. Moreover, these user studies are highly consistent with our quantitative and qualitative results, further indicating the high quality of our dataset.

*Table 3.* Human preference evaluation for image editing.

| Model | FluxKontext | UniWorld-V1 | ShareGPT-4o | Ours (Finetuned) |
|---|---|---|---|---|
| Human Pref. Rate (%) | 16.5 | 17.6 | 30.4 | **35.4** |

### 3.5. Ablation Studies on Per-Module Contributions

To quantify the contribution of each data module, we conducted ablation studies on image generation and editing. The editing analysis is provided in Appendix C.

**Contribution to Generation Capabilities.** Using Harmon

as the baseline, we perform ablation on GenEval (Table 4). The Complex IF and SR modules are key performance drivers. Notably, 8K SR samples improve the Position score from 0.61 to 0.84, demonstrating strong correlation between targeted data and specific capabilities. Additional details on related works, dataset distribution, and data curation are in Appendix A, B, and D, respectively.

*Table 4.* Cumulative ablation for generation capabilities on GenEval. Complex IF denotes Complex Instruction Following and SR denotes Spatial Reasoning in Images. Performance improves as these targeted data modules are added.

| Training Configuration | Two obj. | Count | Position | Color attr. | Overall |
|---|---|---|---|---|---|
| Harmon (Baseline) | 0.86 | 0.66 | 0.74 | 0.48 | 0.76 |
| + Complex IF (6k) | 0.97 | 0.77 | 0.61 | 0.68 | 0.81 |
| + Complex IF + SR (14k) | 0.96 | 0.79 | 0.84 | 0.68 | 0.85 |
| + All Gen. Data (40k) | 0.96 | 0.79 | 0.85 | 0.69 | 0.86 |

## 4. Conclusion

This paper introduces OpenGPT-4o-Image, a comprehensive dataset designed to advance multimodal AI capabilities in image generation and editing through systematic task decomposition and automated data construction. The work addresses significant gaps in existing datasets by providing 80,000 high-quality instruction-image pairs across 11 major domains and 51 subtasks, with particular emphasis on previously underexplored areas such as scientific imagery, complex instruction following, and multi-turn editing. The key contributions include a hierarchical taxonomy that systematically categorizes image generation into five core modules (Style Control, Complex Instruction Following, In-Image Text Rendering, Spatial Reasoning, and Scientific

Imagery) and image editing into six categories with 21 sub-tasks. The automated pipeline leverages GPT-4o to ensure scalable, consistent data generation while maintaining controlled diversity and difficulty distribution. Experimental validation demonstrates the dataset's effectiveness across multiple model architectures and benchmarks. However, the work has limitations. The reliance on GPT-4o for data generation may introduce biases inherent to that model, and the evaluation is primarily conducted on existing benchmarks which may not fully capture real-world application scenarios.

## Acknowledgements

This work was in part supported by the National Natural Science Foundation of China under grants 62472399 and Open Fund of APKL of BIIP, IAI,Hefei Comprehensive National Science Center under grants 24YGXT003.

## Impact Statement

OpenGPT-4o-Image is intended to support non-commercial research on multimodal instruction following, image generation and editing, scientific visualization, creative design, and model evaluation. Because realistic image editing can also enable non-consensual intimate imagery, impersonation, identity fraud, misinformation, fabricated evidence, privacy invasion, and harassment, we release the dataset with explicit safeguards and responsible-use restrictions.

All source data are drawn from public resources with licenses permitting research use or compatible redistribution. Generated instruction-image pairs are released for non-commercial research under CC-BY-NC 4.0, together with terms that prohibit deceptive, malicious, privacy-invasive, illegal, or non-consensual uses. During construction, we filter PII, reject recognizable real human faces, remove sensitive text detected by OCR, and apply prompt- and image-level safety filters for sexual or non-consensual content, violence, hate or harassment, self-harm, illegal acts, and explicitly biased content. These safeguards rejected approximately 8.2% of candidate image-editing pairs.

Dataset-level filtering cannot eliminate all downstream risks. We therefore recommend input/output safety classifiers, watermarking or provenance metadata, identity-consent checks, and human review for high-stakes deployments. The dataset should not be used for surveillance, biometric identification, deception, evidence generation, news manipulation, intimate-content generation, harassment, or other uses that may harm individuals or public trust. Additional details on data sourcing, privacy filtering, safety filtering, and responsible release are provided in Appendix D.2 and Appendix D.3.

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

# A. Related Work

## A.1. Datasets for Image Generation

The advancement of text-to-image generation has been driven by several large-scale datasets (see Table 5 for representative examples). For instance, LAION-Aesthetics-UMAP (dclure, 2022), focus on curating images with high aesthetic scores, while DenseFusion-1M (Li et al., 2024a) provides dense descriptions rich in visual detail. Others, such as Megalith (madeby-ollin, 2024) and Public Domain 12M (Meyer et al., 2024), contribute vast image resources from real-world or public-domain sources. However, often constrained by the capabilities of the models available at the time of their creation, these earlier datasets exhibit limitations in their ability to support complex semantic understanding and precise instruction following. Recently, ShareGPT-4o-Image (Chen et al., 2025a) and Echo-4o-Image (Ye et al., 2025a) have made significant strides in data quality by leveraging the advanced generation capabilities of GPT-4o. While the datasets effectively distill the model's generative potential, its taxonomy is relatively coarse-grained and lacks a systematic, fine-grained structure. This somewhat limits its utility for targeted evaluation and in-depth analysis of specific model capabilities. To address this gap, we introduce OpenGPT-4o-Image, a text-to-image dataset built upon a clear, hierarchical taxonomy. It is designed to provide a resource that combines diversity, practicality, and evaluative depth, enabling more precise training and analysis of a model's multi-dimensional instruction-following abilities.

*Table 5.* **Comparison of Existing Datasets and OpenGPT-4o-Image. MTS** indicates support for multi-turn image editing, **CS** for complex instruction-based image editing, **TES** for visual text-based image editing, **ITR** for In-Image Text Rendering, and **SIG** for Scientific Image Generation.

| Dataset | Size | Types | MTS | CS | TES | ITR | SIG |
|---|---|---|---|---|---|---|---|
| *Image Editing* | | | | | | | |
| Magicbrush (Zhang et al., 2023) | 10K | 5 | ✓ | ✗ | ✗ | - | - |
| Seed-Data-Edit (Ge et al., 2024) | 3.7M | 6 | ✓ | ✗ | ✗ | - | - |
| HQ-Edit (Hui et al., 2024) | 197k | 6 | ✗ | ✗ | ✗ | - | - |
| AnyEdit (Yu et al., 2025) | 2.5M | 25 | ✗ | ✗ | ✗ | - | - |
| IP2P (Brooks et al., 2023) | 313K | 4 | ✗ | ✗ | ✗ | - | - |
| UltraEdit (Zhao et al., 2024) | 4M | 9 | ✗ | ✗ | ✗ | - | - |
| OmniEdit (Wei et al., 2024) | 1.2M | 7 | ✗ | ✗ | ✗ | - | - |
| ImgEdit (Ye et al., 2025b) | 1.2M | 13 | ✓ | ✓ | ✗ | - | - |
| *Image Generation* | | | | | | | |
| text-to-image-2M (jackyhate, 2024) | 2M | - | - | - | - | ✗ | ✗ |
| Laion-aesthetics-umap (dclure, 2022) | 12M | - | - | - | - | ✗ | ✗ |
| Densefusion-1m (Li et al., 2024a) | 1M | - | - | - | - | ✗ | ✗ |
| Journeydb (Pan et al., 2023) | 4M | - | - | - | - | ✗ | ✗ |
| Public Domain 12M (Meyer et al., 2024) | 12M | - | - | - | - | ✗ | ✗ |
| Megalith (madebyollin, 2024) | 10M | - | - | - | - | ✗ | ✗ |
| Text-Render-2M (Chen et al., 2025c) | 2M | - | - | - | - | ✓ | ✗ |
| Echo-4o-Image (Ye et al., 2025a) | 180K | - | - | - | - | ✗ | ✗ |
| *Unify* | | | | | | | |
| ShareGPT-4o-Image (Chen et al., 2025a) | 91K | 29 | ✗ | ✗ | ✗ | ✗ | ✗ |
| OpenGPT-4o-Image | 80K | **51** | ✓ | ✓ | ✓ | ✓ | ✓ |

## A.2. Datasets for Image Editing

Table 5 compares representative instruction-driven image editing datasets (Brooks et al., 2023; Zhang et al., 2023; Ge et al., 2024; Zhao et al., 2024; Hui et al., 2024; Yu et al., 2025; Wei et al., 2024; Ye et al., 2025b; Wang et al., 2025b; Qian et al., 2025). Most of these datasets (Brooks et al., 2023) rely primarily on open-source models to generate edited images. Although MagicBrush (Zhang et al., 2023) and SEED-Data-Edit (Ge et al., 2024) incorporate varying degrees of human

*Table 6.* Comparison of fine-tuning results of different models on our dataset on GEdit-Bench (Liu et al., 2025). ‡ indicates results from our own tests without fine-tuning.

| Model | Background | Color | Material | Motion | Portrait | Style | Add | Remove | Replace | Text | Tone | Avg |
|---|---|---|---|---|---|---|---|---|---|---|---|---|
| *Open-source Models* | | | | | | | | | | | | |
| AnyEdit (Jiang et al., 2025b) | 4.31 | 4.25 | 2.64 | 0.67 | 1.9 | 1.95 | 3.72 | 3.75 | 3.23 | 0.77 | 4.21 | 2.85 |
| IP2P (Brooks et al., 2023) | 3.94 | 5.4 | 3.52 | 1.27 | 2.62 | 4.39 | 3.07 | 1.5 | 3.48 | 1.13 | 5.1 | 3.22 |
| OmniGen (Xiao et al., 2025) | 5.23 | 5.93 | 5.44 | 3.12 | 3.17 | 4.88 | 6.33 | 6.35 | 5.34 | 4.31 | 4.96 | 5.01 |
| Step1X-Edit (Liu et al., 2025) | 7.03 | 6.26 | 6.46 | 3.66 | 5.23 | 7.24 | 7.17 | 6.42 | 7.39 | **7.40** | 6.62 | 6.44 |
| Bagel (de Jong et al., 2006) | 7.44 | 6.99 | 6.26 | 5.09 | 4.82 | 6.04 | **7.94** | 7.37 | 7.31 | 7.16 | 6.17 | 6.60 |
| Bagel-thinking | 7.22 | 7.24 | 6.69 | 7.12 | 6.03 | 6.17 | 7.93 | **7.44** | **7.45** | 3.61 | 6.36 | 6.66 |
| Ovis-U1 (Wang et al., 2025a) | **7.49** | 6.88 | 6.21 | 4.79 | **5.98** | 6.46 | 7.49 | 7.25 | 7.27 | 4.48 | 6.31 | 6.42 |
| OmniGen2 (Wu et al., 2025c) | - | - | - | - | - | - | - | - | - | - | - | 6.42 |
| Step1X-Edit (v1.1) | 7.45 | **7.38** | **6.95** | **4.73** | 4.70 | 7.11 | 8.2 | 7.59 | 7.8 | 7.91 | 6.85 | **6.97** |
| FluxKontext dev (Labs et al., 2025) | 7.06 | 7.03 | 5.52 | 5.62 | 4.68 | 5.55 | 6.95 | 6.76 | 6.13 | 6.10 | 7.48 | 6.26 |
| MagicBrush (Zhang et al., 2023) | 6.17 | 5.41 | 4.75 | 1.55 | 2.9 | 4.1 | 5.53 | 4.13 | 5.1 | 1.33 | 5.07 | 4.19 |
| UniWorld-V1 (Lin et al., 2025) | 4.92 | 6.37 | 4.79 | 1.85 | 4.03 | 5.64 | 7.23 | 6.17 | 5.70 | 1.15 | 5.54 | 4.85 |
| OmniGen‡ (Xiao et al., 2025) | 4.87 | 5.57 | 4.75 | 2.57 | 4.09 | 5.84 | 6.04 | 4.77 | 5.42 | 4.41 | 5.21 | 4.87 |
| OmniGen2‡ (Wu et al., 2025c) | 7.04 | 6.32 | 6.21 | 3.56 | 2.94 | 6.74 | 6.42 | 6.14 | 6.93 | 4.86 | 6.62 | 5.80 |
| *Proprietary Models* | | | | | | | | | | | | |
| Doubao (Shi et al., 2024) | 8.07 | 7.36 | 7.20 | 5.38 | 6.28 | 7.2 | 8.05 | 7.71 | 7.87 | 4.01 | 7.67 | 6.98 |
| GPT-4o (Open, 2025) | 6.96 | 6.85 | 7.10 | 5.41 | 6.74 | 7.44 | 7.51 | 8.73 | 8.55 | 8.45 | 8.69 | 7.49 |
| *Finetuning* | | | | | | | | | | | | |
| MagicBrush | 5.84 | 6.07 | 5.08 | 3.41 | 4.17 | 5.94 | 5.84 | 5.95 | 5.01 | 2.41 | 6.36 | 5.10 |
| OmniGen | 5.83 | 6.79 | 5.25 | 4.82 | 4.64 | 5.83 | 6.08 | 5.66 | 6.02 | 3.90 | 6.24 | 5.55 |
| OmniGen2 | 7.23 | 6.46 | 6.73 | 4.65 | 4.81 | 7.07 | 6.69 | 6.42 | 7.03 | 5.43 | 6.83 | 6.31 |
| UniWorld-V1 | 5.38 | **7.38** | 5.22 | 3.52 | 4.03 | **6.88** | 7.07 | 5.23 | 5.50 | 2.13 | **7.39** | 5.43 |

quality control, their limited ability to fully interpret editing instructions often leads to suboptimal results. Specifically, InstructPix2Pix (IP2P) employs GPT-3 (Floridi &Chiriatti, 2020) to generate editing instructions and P2P (Hertz et al., 2022) for image editing; MagicBrush leverages extensive human annotations to improve data fidelity; HQ-Edit (Hui et al., 2024) uses DALL·E to produce paired images, but the outputs lack realism; AnyEdit (Yu et al., 2025) and OmniEdit (Wei et al., 2024) design alternative pipelines to support diverse editing tasks; and UltraEdit(Zhao et al., 2024) introduces a large-scale region-based editing dataset. In addition, ShareGPT-4o-Image (Chen et al., 2025a) constructs 46K instruction-driven image editing pairs using GPT-4o (Hurst et al., 2024), but it fails to adequately cover a broad spectrum of editing types, such as reference image editing. In contrast, we introduce OpenGPT-4o-Image, which selects high-quality source images, designs a comprehensive taxonomy of editing categories, and leverages GPT-4o to generate diverse instructions and high-quality edited images.

*Table 7.* Data scaling results on GEdit-Bench and ImgEdit-Bench, obtained by randomly sampling datasets of different sizes. Two Avg represents the average performance on GEdit-Bench together with the overall average on ImgEdit-Bench.

| Size | Background | Color | Material | Motion | Portrait | Style | Add | Remove | Replace | Text | Tone | Avg |
|---|---|---|---|---|---|---|---|---|---|---|---|---|
| *GEdit-Bench* | | | | | | | | | | | | |
| 20K | 5.60 | 7.30 | 6.46 | 3.82 | 3.77 | 6.57 | 7.73 | 7.73 | 5.60 | 1.90 | 6.83 | 5.50 |
| 30K | 6.80 | 7.64 | 6.23 | 4.98 | 3.19 | 6.84 | 7.66 | 5.52 | 5.45 | 1.13 | 6.87 | 5.79 |
| 40K | 5.38 | 7.38 | 5.22 | 3.52 | 4.03 | 6.88 | 7.07 | 5.23 | 5.50 | 2.13 | 7.39 | 5.43 |

| Size | Add | Adjust | Extract | Replace | Remove | Background | Style | Hybrid | Action | Overall | Two Avg |
|---|---|---|---|---|---|---|---|---|---|---|---|
| *ImgEdit-Bench* | | | | | | | | | | | |
| 20K | 4.09 | 4.28 | 2.46 | 3.85 | 2.93 | 3.99 | 4.58 | 3.19 | 4.07 | 3.72 | 6.46 |
| 30K | 4.16 | 4.02 | 2.53 | 3.68 | 2.75 | 4.06 | 4.49 | 2.98 | 3.81 | 3.64 | 6.53 |
| 40K | 4.34 | 4.28 | 2.66 | 3.92 | 3.30 | 4.15 | 4.62 | 3.43 | 3.97 | 3.86 | 6.58 |

*Table 8.* Comparison of fine-tuning results of different models on our dataset on DPG-Bench (Hu et al., 2024). ‡ indicates results from our own tests without fine-tuning.

| Method | Global | Entity | Attribute | Relation | Other | Overall |
|---|---|---|---|---|---|---|
| *Open-source Models* | | | | | | |
| SDXL (Podell et al., 2023) | 83.27 | 82.43 | 80.91 | 86.76 | 80.41 | 74.65 |
| Hunyuan-DiT (Li et al., 2024b) | 84.59 | 80.59 | 88.01 | 74.36 | 86.41 | 78.87 |
| DALLE3 (OpenAI, 2024) | 90.97 | 89.61 | 88.39 | 90.58 | 89.83 | 83.50 |
| SD3-medium (AI, 2024) | 87.90 | **91.01** | 88.83 | 80.70 | 88.68 | 84.08 |
| FLUX.1-dev (Labs, 2024) | 82.1 | 89.5 | 88.7 | 91.1 | 89.4 | 84.0 |
| Show-o (Xie et al., 2024) | 79.33 | 75.44 | 78.02 | 84.45 | 60.80 | 67.27 |
| EMU3 (Wang et al., 2024) | 85.21 | 86.68 | 86.84 | 90.22 | 83.15 | 80.60 |
| TokenFlow-XL (Qu et al., 2025) | 78.72 | 79.22 | 81.29 | 85.22 | 71.20 | 73.38 |
| Janus Pro (Chen et al., 2025d) | 86.90 | 88.90 | 89.40 | 89.32 | 89.48 | 84.19 |
| T2I-R1 (Jiang et al., 2025a) | 91.79 | 90.23 | 89.05 | 90.13 | 89.48 | 84.76 |
| BLIP3-o 4B (Chen et al., 2025b) | - | - | - | - | - | 79.36 |
| BLIP3-o 8B (Chen et al., 2025b) | - | - | - | - | - | 81.60 |
| BAGEL (Deng et al., 2025) | 88.94 | 90.37 | **91.29** | 90.82 | 88.67 | 85.07 |
| UniWorld-V1 (Lin et al., 2025) | 83.64 | 88.39 | 88.44 | 89.27 | 87.22 | 81.38 |
| Harmon (Wu et al., 2025d) | 83.53 | 85.45 | 86.31 | 87.93 | 89.03 | 81.27 |
| OmniGen† (Xiao et al., 2025) | 87.90 | 88.97 | 88.47 | 87.95 | 83.56 | 81.16 |
| OmniGen2‡ (Wu et al., 2025c) | 85.69 | 88.98 | 86.88 | **91.54** | **91.38** | 82.37 |
| *Finetuning* | | | | | | |
| OmniGen | 86.69 | 88.08 | 90.40 | 91.39 | 90.77 | 83.76 |
| OmniGen2 | 88.75 | 89.31 | 89.55 | 90.67 | 91.08 | 84.01 |
| UniWorld-V1 | **92.81** | 89.56 | 89.41 | 88.93 | 88.52 | 83.66 |
| Harmon | 92.30 | 90.35 | 90.45 | 91.47 | 91.19 | **85.60** |

# B. Dataset Distribution

## B.1. Distribution of Image Editing Data

In this section, we provide a more detailed characterization of the distribution of the image editing dataset. As summarized in Table 14, the corpus is organized into six major categories: Subject Manipulation (19k), Text Editing (3k), Complex Instruction Editing (4k), Multi-turn Editing (1.5k), Global Editing (5k), and Other Challenging Editing (8k). Because Subject Manipulation and Global Editing represent practical and canonical editing tasks, we allocate comparatively larger quantities to these categories to strengthen performance in these areas. For Text Editing, Multi-turn Editing, and the sub-tasks under Other Challenging Editing, we employ templated generation to curate a relatively smaller but targeted set of instances, enabling an examination of how high-quality fine-tuning translates into improvements for these capabilities. Finally, we include a moderate amount of Complex Instruction Editing to further enhance instruction following; this capability shows substantial gains after fine-tuning, as reflected by marked improvements on ImgEdit-Bench.

## B.2. Distribution of Image Generation Data

Similarly, the generation dataset is strategically structured to build a comprehensive range of creative and technical capabilities, with its distribution detailed in Table 15. The corpus is organized into five major categories: Style Control (13k), Scientific Imagery (10k), Spatial Reasoning (8k), Complex Instruction Following (6k), and In-Image Text Rendering (3k). The largest allocations are given to Style Control and Scientific Imagery to, respectively, master a foundational pillar of creative expression and establish a robust data foundation for professional domains where high-quality training data is typically scarce. Substantial resources are also devoted to Spatial Reasoning and Complex Instruction Following to systematically enhance the model's grasp of logical relationships, geometric precision, and compositional directives. Finally, a more targeted collection for In-Image Text Rendering (3k) is included to specifically address persistent challenges in

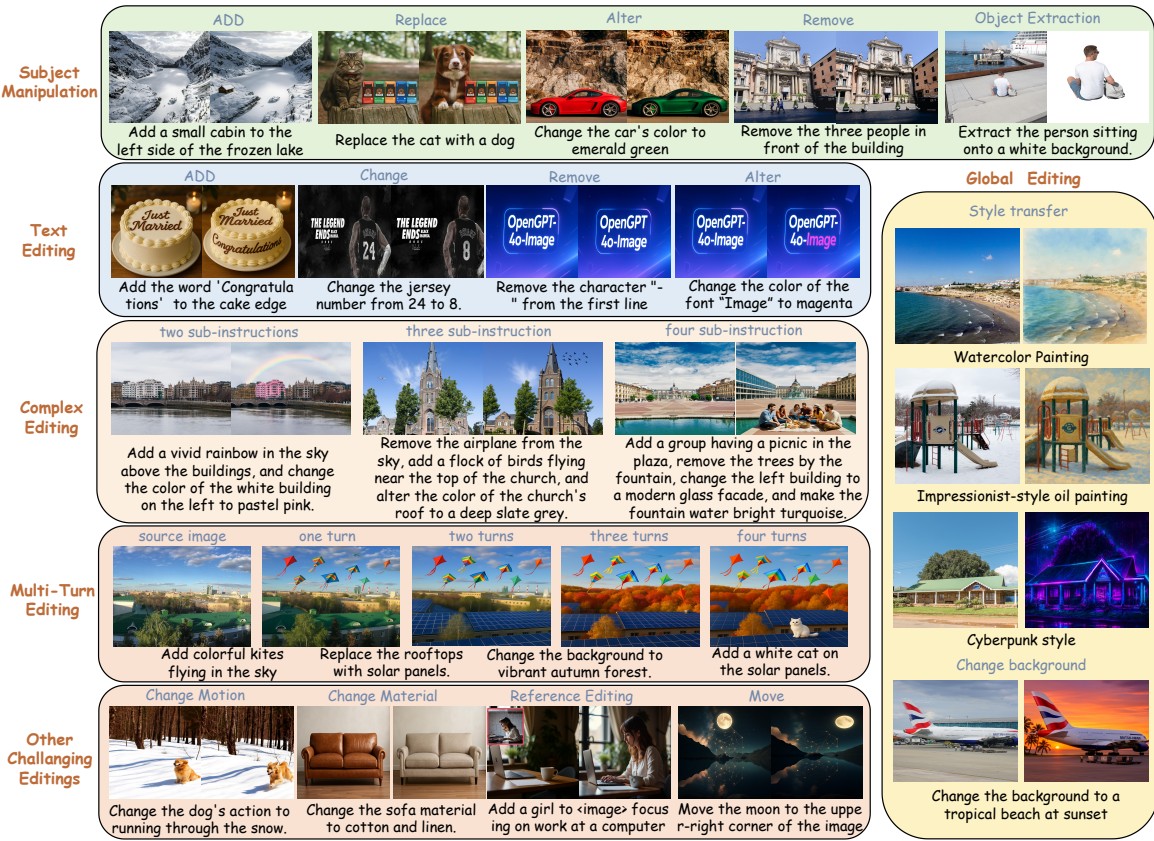

*Figure 4.* **Illustrative examples of image editing from OpenGPT-4o-Image.** We comprehensively categorize image editing tasks into six groups: (a) Subject Manipulation, focusing on region-based editing; (b) Text Editing, involving modifications to textual content embedded in images; (c) Complex Editing, involving the combination of multiple simple editing instructions; (d) Multi-Turn Editing, consisting of iterative, multi-round editing interactions; (e) Other Challenging Editings, covering additional difficult editing scenarios; and (f) Global Editing, which targets holistic modifications across the entire image.

textual accuracy and typography, aiming to improve the model's reliability in this critical area.

## C. Extended Experiments

### C.1. Supplementary Quantitative Experiments

To provide a fuller account of the evaluation, we partition the additional experiments into three facets: scaling the training data, assessing benchmark outcomes, and comparing unified dataset. Table 7 reports results obtained by fine-tuning UniWorld-V1 on incrementally larger samples drawn from our dataset; the measurements consistently indicate performance growth with increasing data volume, reinforcing the importance of dataset size in enabling stronger generalization.

In parallel, Table 6 presents a comprehensive evaluation on GEdit-Bench, which demonstrates that UniWorld-V1 attains state-of-the-art results among closed-source systems on representative editing tasks, including color change and tone transfer. Similarly, on the DPG-Bench, the fine-tuned Harmon model achieves the best performance, as shown in Table 8. Moreover, the same fine-tuning protocol yields marked improvements for the remaining models considered, suggesting that the gains are not confined to specific models or benchmarks but instead reflect stronger generalization capacity. In terms of the unified dataset, we conducted a comparison between our dataset and ShareGPT4o. Specifically, we employed UniWorld-V1 distributions to finetune both the generation and editing components. As summarized in Table 9, our dataset consistently outperforms ShareGPT4o in both generation and editing tasks. This advantage can be attributed to our meticulous categorization, the diversity of editing instructions, and the high-quality editing outcomes.

*Table 9.* Comparison of fine-tuning results on UniWorld-V1 with ShareGPT-4o-Image. Two Avg represents the average performance on GEdit-Bench together with the overall average on ImgEdit-Bench.

| Dataset | Background | Color | Material | Motion | Portrait | Style | Add | Remove | Replace | Text | Tone | Avg |
|---|---|---|---|---|---|---|---|---|---|---|---|---|
| *GEdit-Bench* | | | | | | | | | | | | |
| ShareGPT-4o-Image | 4.87 | 7.70 | 5.59 | 2.20 | 4.04 | 7.29 | 6.94 | 5.03 | 5.13 | 2.12 | 6.93 | 5.26 |
| Ours | 5.38 | 7.38 | 5.22 | 3.52 | 4.03 | 6.88 | 7.07 | 5.23 | 5.50 | 2.13 | 7.39 | 5.43 |

| Dataset | Add | Adjust | Extract | Replace | Remove | Background | Style | Hybrid | Action | Overall | Two Avg |
|---|---|---|---|---|---|---|---|---|---|---|---|
| *ImgEdit-Bench* | | | | | | | | | | | |
| ShareGPT-4o-Image | 4.16 | 4.24 | 2.45 | 3.85 | 2.9 | 3.98 | 4.65 | 2.99 | 3.77 | 3.7 | 6.33 |
| Ours | 4.34 | 4.28 | 2.66 | 3.92 | 3.3 | 4.15 | 4.62 | 3.43 | 3.97 | 3.86 | 6.58 |

| Dataset | Single object | Two object | Counting | Colors | Position | Color attribution | Overall |
|---|---|---|---|---|---|---|---|
| *GenEval-Bench* | | | | | | | |
| ShareGPT-4o-Image | 0.99 | 0.94 | 0.82 | 0.87 | 0.56 | 0.71 | 0.82 |
| Ours | 0.99 | 0.96 | 0.82 | 0.88 | 0.60 | 0.73 | 0.83 |

| Size | Global | Entity | Attribute | Relation | Other | Overall |
|---|---|---|---|---|---|---|
| *DPG-Bench* | | | | | | |
| ShareGPT-4o-Image | 87.46 | 88.75 | 88.97 | 90.36 | 88.76 | 82.71 |
| Ours | 92.81 | 89.56 | 89.41 | 88.93 | 88.52 | 83.66 |

## C.2. Supplementary Qualitative Experiments

Qualitative results for generation are shown in Figure 5, with analysis in Section 3.3. For the editing task, beyond the qualitative example in Figure 6 (which shows the model simultaneously removing a laptop and adding a light blue sofa), we also present a comprehensive quantitative study. This study, covering multiple editing types like Add Subject and Change Background, indicates that fine-tuning improves not only editing performance but also image fidelity. Moreover, for hybrid instructions, our dataset enhances the model's ability to follow complex, compositional directives.

## C.3. Supplementary Quantitative Experiments on Unified dataset

We further compare unified training with separate training for generation and editing. As shown in Table 12, for image editing, separate fine-tuning surpasses unified training on ImgEdit-Bench, whereas unified training outperforms separate fine-tuning on GEdit-Bench. For image generation, separate fine-tuning performs on par with unified training on GenEval-Bench, but surpasses it on DPG-Bench. These results suggest a benchmark-dependent trade-off between unified and task-specific training, indicating potential task interference and underscoring the importance of tailored optimization for different evaluation regimes.

## C.4. Human Preference Evaluation for Image Generation

Similar to the image editing evaluation in Section 3.4, we conducted a human preference study for image generation. The same 20 expert evaluators were asked to select the best result from three models on 200 generation samples: UniWorld-V1, ShareGPT-4o-Image, and our finetuned UniWorld-V1. As shown in Table 10, our fine-tuned model achieves the highest preference rate, demonstrating consistent improvements in human-perceived generation quality.

## C.5. Data Consistency Optimization and Quality Verification

We acknowledge that utilizing generative models (e.g., GPT-4o) for dataset creation may introduce challenges regarding identity (ID) and background consistency. To address concerns about the dataset's fidelity and its long-term relevance for fine-tuning evolving models, we implemented a rigorous prompting strategy. Specifically, we incorporated an explicit constraint—"Keep the object ID consistent and other non-edited areas unchanged"—into the generation pipeline. To validate this approach, we conducted a human expert evaluation on a randomly sampled 20% subset, which confirmed significant improvements in preserving both ID and background integrity. Furthermore, quantitative assessment using EditScore demonstrated the reliability of our dataset, achieving scores of 8.82 in visual consistency and 9.46 in instruction adherence. These results attest to the high quality of the generated data, supporting its effectiveness as a robust training resource.

*Table 10.* Human preference evaluation for image generation.

| Model | UniWorld-V1 | ShareGPT-4o | **Ours (Finetuned)** |
|---|---|---|---|
| **Human Pref. Rate (%)** | 19.5 | 37.2 | **43.3** |

### C.6. Contribution to Editing Capabilities.

As shown in Table 11, we conduct detailed ablation studies and observe that atomic modules such as Remove and Replace precisely improve their corresponding operations, while the Complex module enhances compositional abilities. This further validates the effectiveness of different subsets of our dataset.

*Table 11.* Cumulative ablation for editing capabilities on ImgEdit-Bench.

| Training Configuration | Add | Replace | Remove | Hybrid | Style | **Overall** |
|---|---|---|---|---|---|---|
| OmniGen2 (Baseline) | 3.60 | 3.81 | 2.54 | 2.71 | 4.57 | 3.39 |
| + Remove (7k) | 4.16 | 3.92 | 3.78 | 2.75 | 4.53 | 3.56 |
| + Remove + Replace (11k) | 4.15 | 4.21 | 3.62 | 2.82 | 4.59 | 3.62 |
| + Remove + Replace + Complex (15k) | 4.13 | 4.16 | 3.71 | 3.12 | 4.55 | 3.70 |
| + All Edit. Data (40k) | 4.15 | 4.11 | 3.64 | 2.99 | 4.69 | 3.82 |

## D. Discussion

### D.1. Data Curation and Quality Control Strategy

A primary challenge in curating our dataset is ensuring high fidelity to complex, compositional instructions. While frontier models like GPT-4o consistently produce aesthetically pleasing images, their instruction-following capabilities often degrade when faced with highly complex prompts demanding adherence to numerous fine-grained details. This limitation is also prevalent in existing open-source models. Consequently, a simple post-hoc filtering strategy proved ineffective, as high aesthetic quality is often a poor proxy for semantic correctness and instruction-following accuracy.

To address this, we adopted a proactive quality control strategy centered on meticulous, fine-grained data curation prior to generation. Our methodology is guided by two core principles:

**Hierarchical Categorization.** We first establish a clear hierarchy by defining distinct modules (e.g., Style Control, Spatial Reasoning) and then partitioning them into more granular, well-defined sub-classes. This ensures thematic coherence and targeted data collection.

**Difficulty Calibration.** Within each category, we carefully calibrate the task difficulty to occupy a specific "sweet spot". The instructions are designed to be challenging for current open-source models yet demonstrably solvable by a state-of-the-art model like GPT-4o.

**Instruction Recalibration via Post-Processing** Despite the advanced capabilities of models like GPT-4o, semantic discrepancies between the input prompt and the generated image—such as numerical inconsistencies (e.g., generating three objects instead of the requested four)—can still occur. To address this, we adopted a pragmatic philosophy: "There are no invalid images, only invalid text." Instead of discarding these mismatched samples, we implemented a reverse-alignment strategy. Specifically, we recalibrate the textual instructions to accurately describe the actual visual content of the generated image. This approach not only ensures strict alignment between the instruction and the image but also maximizes data utilization by converting potential "failures" into high-quality, valid training pairs.

This principled approach ensures that our dataset is not only of high visual quality but is also semantically accurate, posing a meaningful and well-defined challenge for advancing model capabilities.

### D.2. Ethics, Privacy, and Safety

**Ethical standards.** Our collection and release follow the ICML and NeurIPS Codes of Ethics. We respect source licenses, avoid releasing data that identifies private individuals with reasonable effort, document intended and prohibited uses, and explicitly address misuse risks.

*Table 12.* Comparison of fine-tuning performance on UniWorld-V1 across different training strategies. The table evaluates model performance when fine-tuned separately on generation-only and editing-only datasets versus a unified dataset combining both.

| Dataset | Background | Color | Material | Motion | Portrait | Style | Add | Remove | Replace | Text | Tone | Avg |
|---|---|---|---|---|---|---|---|---|---|---|---|---|
| | | | | | *GEdit-Bench* | | | | | | | |
| Edit-Only | 5.38 | 7.38 | 5.22 | 3.52 | 4.03 | 6.88 | 7.07 | 5.23 | 5.50 | 2.13 | 7.39 | 5.43 |
| Unify | 6.01 | 7.47 | 6.97 | 2.37 | 4.54 | 7.05 | 7.11 | 5.99 | 5.36 | 2.01 | 6.95 | 5.62 |

| Dataset | Add | Adjust | Extract | Replace | Remove | Background | Style | Hybrid | Action | Overall |
|---|---|---|---|---|---|---|---|---|---|---|
| | | | | | *ImgEdit-Bench* | | | | | |
| Edit-Only | 4.34 | 4.28 | 2.66 | 3.92 | 3.30 | 4.15 | 4.62 | 3.43 | 3.97 | 3.86 |
| Unify | 4.13 | 3.86 | 2.45 | 3.52 | 2.81 | 3.86 | 4.65 | 3.32 | 3.78 | 3.60 |

| Dataset | Single object | Two object | Counting | Colors | Position | Color attribution | Overall |
|---|---|---|---|---|---|---|---|
| | | | *GenEval-Bench* | | | | |
| Gen-Only | 0.99 | 0.96 | 0.82 | 0.88 | 0.60 | 0.73 | 0.83 |
| Unify | 0.99 | 0.96 | 0.80 | 0.87 | 0.60 | 0.72 | 0.83 |

| Dataset | Global | Entity | Attribute | Relation | Other | Overall |
|---|---|---|---|---|---|---|
| | | | *DPG-Bench* | | | |
| Gen-Only | 92.81 | 89.56 | 89.41 | 88.93 | 88.52 | 83.66 |
| Unify | 89.62 | 89.39 | 88.80 | 87.66 | 88.39 | 82.45 |

**Source data and license filtering.** The source corpus combines public image/editing datasets, GPT-4o generated images, and curated high-resolution public images. For editing, we use SEED-Data-Edit, ImgEdit, OmniEdit, GPT-4o generated images, and curated public images; for public image sources, we use resources such as COCO, OpenImages/Unsplash-derived data, and other public images only when the license permits research use or compatible redistribution. We accept sources under CC-BY, CC-BY-NC, Apache-2.0, MIT, Unsplash-compatible, or explicit research-use terms, and reject sources with missing or ambiguous license, redistribution, derivative-use, or research-use status. Attribution, non-commercial, and downstream-use requirements are preserved in the release metadata. Newly generated instruction-image pairs are released for non-commercial research under CC-BY-NC 4.0 with responsible-use terms prohibiting deceptive, malicious, privacy-invasive, illegal, or non-consensual applications.

**PII definition.** We define PII as visual or textual information that can reasonably identify a specific person, including recognizable real human faces, names, phone numbers, email or physical addresses, license plates, personal documents, social media handles, signatures, and other unique identifiers. Recognizable real faces are treated as potential PII even without names. Our prompt templates avoid named private individuals, public figures, and unique personal identifiers.

**PII filtering protocol.** We apply privacy filtering before release. A RetinaFace-based detector scans source, generated, and edited images, and samples containing clear real human faces are rejected. OCR and pattern/classifier checks remove sensitive text such as phone numbers, addresses, IDs, license plates, and email addresses. Ambiguous cases are conservatively removed or manually reviewed. Retained human-like figures must not identify a specific individual with reasonable effort, and the release documentation reports removal counts for each privacy filter.

**Safety taxonomy.** Unsafe content includes sexual or non-consensual intimate imagery, violence or gore, hate or harassment, demeaning stereotypes, self-harm, illegal acts, dangerous instructions, exploitation of minors, privacy-invasive or identity-deceptive content, deceptive-document or evidence fabrication, and explicitly biased or discriminatory content. This taxonomy is designed to cover the harmful applications raised in the ethics review, including non-consensual intimate imagery, identity fraud, misinformation, and privacy invasion.

**Safety filtering protocol.** We filter at three stages: prompt templates that request unsafe or privacy-invasive edits are rejected; generation-time API safeguards refuse unsafe requests; and post-generation checks use OpenAI's Content Moderation API plus a custom vision-language safety classifier. A sample is removed if any stage flags it under the taxonomy

*Table 13.* Source repositories and license handling used during data collection.

| Source | Role in our corpus | License handling |
|---|---|---|
| SEED-Data-Edit | Editing source pairs and multi-turn examples | CC-BY-NC-4.0; retained for non-commercial research use with attribution and source-license metadata. |
| ImgEdit | High-quality editing source pairs and benchmark-aligned tasks | Apache-2.0; retained with attribution and original license metadata. |
| OmniEdit-Filtered-1.2M | Reference and subject-driven editing source pairs | MIT; retained with attribution and original license metadata. |
| COCO, OpenImages, Unsplash-derived and curated public images | Public source images for generation/editing tasks | Accepted only when image-level license or source terms permit research use or compatible redistribution; ambiguous-license records are excluded. |
| GPT-4o generated images | Synthetic source and target images | Released as derived instruction-image pairs under CC-BY-NC 4.0 plus responsible-use restrictions. |

above, with ambiguous cases handled conservatively. After filtering, we retained 40.5k image-editing pairs (Table 13); the 8.2% rejection rate therefore corresponds to approximately 44.1k candidate pairs and 3.6k rejected pairs. Here, "safe" means screened against these explicit categories, not free of all downstream misuse risk.

**Bias and diversity audit.** GPT-4o may propagate model-specific biases and cultural overrepresentation. We mitigate this risk through a taxonomy that includes non-Western art traditions, multilingual text rendering, scientific imagery, and diverse subject and scene pools. The dataset is not designed to identify people , we remove recognizable real faces as potential PII and document this residual bias risk and recommend that downstream users inspect aggregate properties relevant to their target deployment, such as language/script coverage, geographic or cultural references, and the presence of human-like subjects, and apply filtering or re-weighting when a particular balance is required.

### D.3. Broader Impact and Responsible Release

**Intended uses.** OpenGPT-4o-Image is intended for non-commercial research on multimodal instruction following, image generation/editing, scientific and educational visualization, model evaluation, and dataset construction methodology.

**Unintended and prohibited uses.** The dataset must not be used to generate non-consensual intimate imagery, impersonate real people, fabricate evidence, manipulate news or civic information, invade privacy, perform biometric identification, harass or discriminate against individuals or groups, create forged documents, bypass platform safety systems, or support illegal activity. These restrictions address known risks of misinformation, privacy violation, identity fraud, and the "liar's dividend" from synthetic media (Chesney &Citron, 2019).

**Downstream safeguards.** Dataset-level filtering cannot control deployment. Downstream users should add input/output safety classifiers, identity-consent checks, watermarking, provenance metadata such as C2PA-style records, synthetic-content disclosure, abuse reporting, and human review for high-stakes settings. Models trained on this dataset should not be deployed in open-ended public editing systems without such safeguards.

**Residual risks and limitations.** Automated face detection, OCR, and safety classifiers can produce false negatives or false positives, especially across languages, cultures, image styles, and low-quality or stylized imagery. GPT-4o is a black-box generator, so the dataset may inherit unknown model biases and exact pixel-level regeneration is not guaranteed. We release the static dataset and construction taxonomy to support reproducibility. Our claims are limited to the quality, semantic alignment, and safety categories that we explicitly measure or filter, and should not be read as a guarantee of unrestricted deployment safety.

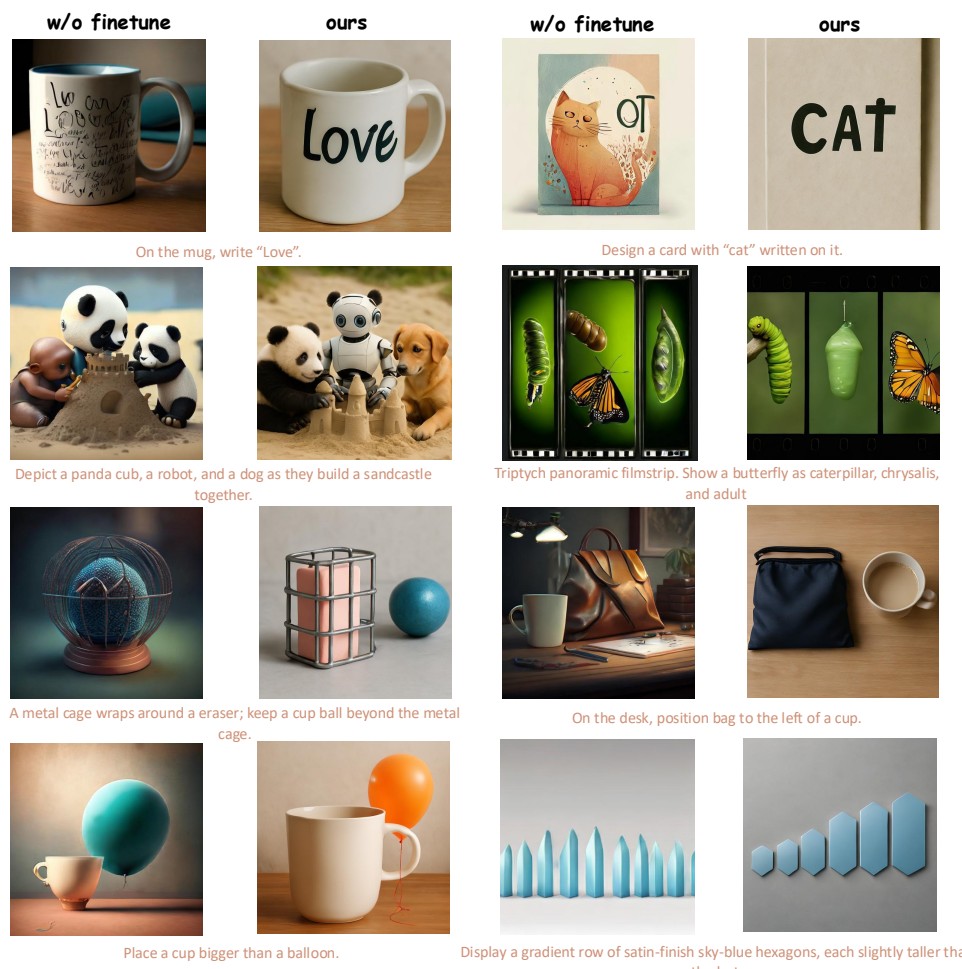

*Figure 5.* Qualitative comparison of Harmon before and after fine-tuning on our dataset.

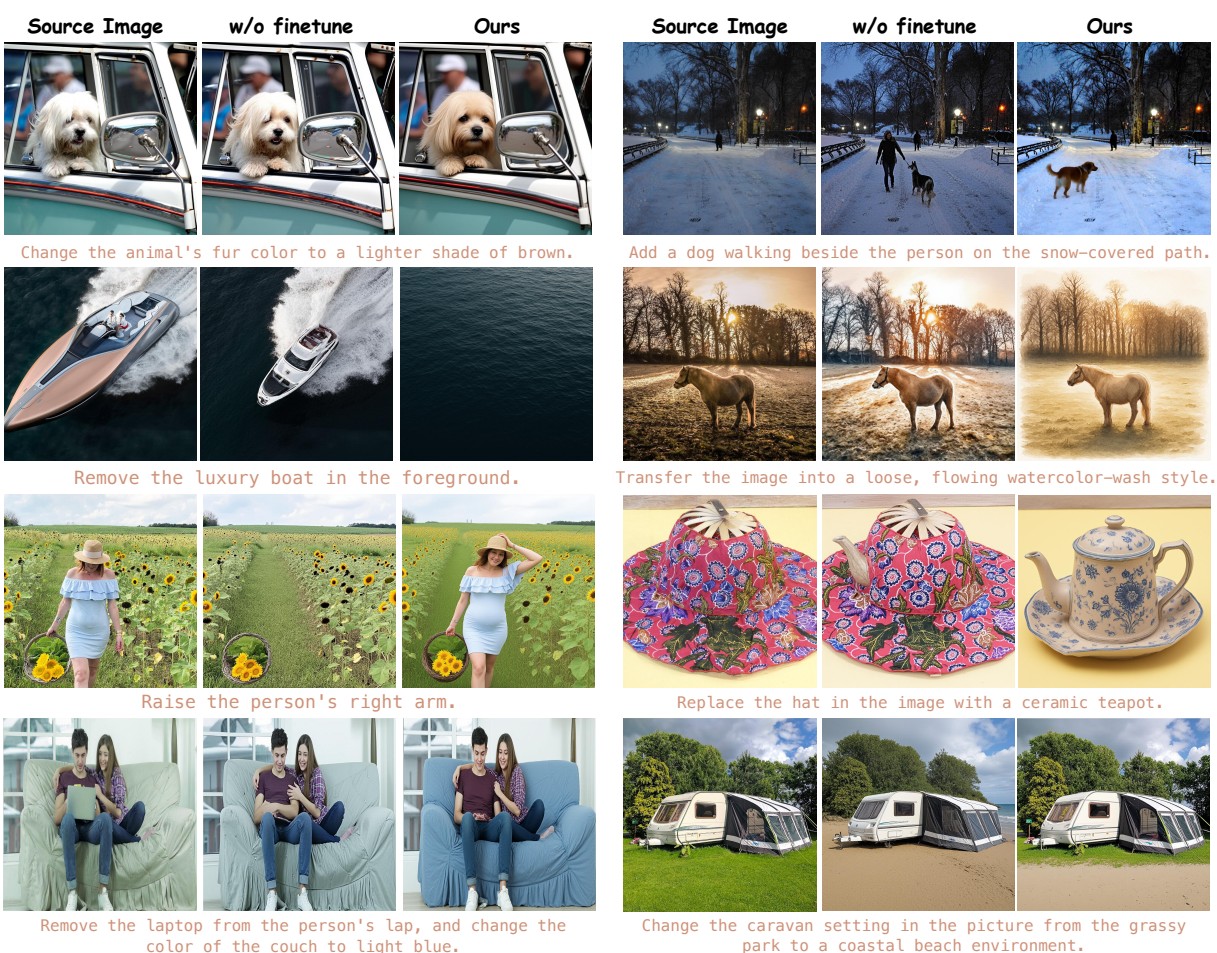

*Figure 6.* Qualitative comparison of UniWorld-V1 before and after fine-tuning on our dataset.

*Table 14.* Overview of the image editing dataset, including task categories, number of instances, and corresponding definitions.

| Task | Number | Definition |
|---|---|---|
| **Subject Manipulation (19k)** | | |
| Add | 4k | Add introduces a new element into the source image. |
| Replace | 4.2k | Replace substitutes an object in the image with a different object. |
| Alter | 1.8k | Alter refers to modifying an existing object's attributes. |
| Remove | 7.2k | Remove refers to eliminating an existing object from the image. |
| Obj Extraction | 1.8k | Object extraction isolates and extracts a specific object from an image. |
| **Text Editing (3k)** | | |
| Text Add | 750 | Text Add inserts textual elements into an image. |
| Text Replace | 750 | Text Replace substitutes a textual element in the image with a new one. |
| Text Alter | 750 | Text Alter modifies the attributes of an textual element (e.g., color). |
| Text Remove | 750 | Text Remove eliminates an existing textual element from an image. |
| **Complex Instruction Editing (4k)** | | |
| Sub-Ins 2 | 1k | A complex instruction consists of two simple editing operations. |
| Sub-Ins 3 | 2k | A complex instruction consists of three simple editing operations. |
| Sub-Ins 4 | 1k | A complex instruction consists of four simple editing operations. |
| **Multi-turn Editing (1.5k)** | | |
| 2 Turns | 500 | A multi-turn editing operation consists of two simple editing rounds. |
| 3 Turns | 500 | A multi-turn editing operation consists of three simple editing rounds. |
| 4 Turns | 500 | A multi-turn editing operation consists of four simple editing rounds. |
| **Global Editing (5k)** | | |
| Change BG | 3.2k | Background replacement refers to substituting the surrounding environment of the subject. |
| Style Transfer | 1.8k | Style transfer refers to the process of modifying the style of an image according to given instructions. |
| **Other Challenging Editing (8k)** | | |
| Ref Image | 3.5k | Reference image editing adds specified subjects into the source image. |
| Change Motion | 2k | Motion modification alters the expressions and movements of objects. |
| Change Material | 2k | Material transformation modifies the texture of clothing. |
| Obj Movement | 500 | Object movement refers to moving an object from one location to another within the image. |

*Table 15.* Overview of the image generation dataset, including task categories, number of instances, and corresponding definitions.

| Task | Number | Definition |
|---|---|---|
| **Style Control (13k)** | | |
| Artistic Traditions | 3.5k | Renders historical and cultural art styles. |
| Media and Illustration | 4.5k | Renders aesthetics from media and illustration. |
| Photographic Styles | 3k | Emulates various photographic techniques and moods. |
| Speculative and Fantasy Styles | 2k | Creates speculative and fantasy genre aesthetics. |
| **Complex Instruction Following (6k)** | | |
| Multi-Attribute Combination | 500 | Applies multiple attributes to subjects. |
| Multi-Subject Interaction and Action | 500 | Depicts interactions between multiple subjects. |
| Complex Spatial Composition | 750 | Arranges elements in complex spatial layouts. |
| Temporal Sequence Coherence | 500 | Generates a logical sequence of events. |
| Action Trajectory Rendering | 750 | Renders the trajectory of moving objects. |
| Causal Reasoning | 3k | Depicts cause-and-effect relationships. |
| **In-Image Text Rendering (3k)** | | |
| Textual Accuracy | 500 | Renders text verbatim from the prompt. |
| Typography | 500 | Controls text font, style, and appearance. |
| Structured Text Layout | 500 | Arranges text in structured layouts (e.g., multi-line). |
| Text-Graphic Integration | 500 | Integrates text coherently with image graphics. |
| Multilingual Support | 500 | Renders text in non-English languages. |
| Textual Tone and Style | 500 | Aligns text style with the image's aesthetic. |
| **Spatial Reasoning in Images (8k)** | | |
| Containment | 2k | Depicts one object inside another. |
| Relative Position | 2k | Places objects in specified relative positions. |
| Comparative Reasoning | 2k | Compares object attributes like size or color. |
| Symmetry Analysis | 500 | Generates symmetrical object arrangements. |
| Size Reasoning | 500 | Renders objects with correct relative sizes. |
| Object Counting | 1k | Generates a specific number of objects. |
| **Scientific Imagery (10k)** | | |
| Mathematics | 1k | Visualizes mathematical concepts. |
| Ecology | 1k | Creates imagery of ecosystems and species. |
| Astronomy | 1k | Generates images of celestial bodies and phenomena. |
| Biological | 1.2k | Illustrates biological structures and organisms. |
| Culture and History | 2.2k | Depicts historical events and cultural artifacts. |
| Earth Science | 1.4k | Visualizes geological and weather phenomena. |
| Mechanical Engineering | 1.2k | Renders mechanical engineering diagrams and systems. |
| Physics | 1.2k | Illustrates physical laws and concepts. |

