# OpenReview forum: "OpenGPT-4o-Image: A Comprehensive Dataset for Advanced Image Generation and Editing"
_ICML.cc/2026/Conference — ICML 2026 regular_

### Official Review · Reviewer_1txg · 2026-03-08

**Soundness:** 3
**Presentation:** 3
**Significance:** 3
**Originality:** 3
**Overall Recommendation:** 5
**Confidence:** 4

**Summary:**

The authors intend to focus on a relevant issue that unified multimodal models for image generation and editing are restricted by training data that lacks systematic structure and complex real-world scenarios. Overall, the article's fundamental concept is to construct a large-scale, hierarchically categorized dataset through an automated pipeline powered by GPT-4o, covering both basic and challenging tasks like scientific imagery and complex instruction editing. This work contributes the 80k instruction-image OpenGPT-4o-Image dataset across 11 major domains and 51 subtasks, and extensive experiments validate that fine-tuning models on this dataset brings notable performance improvements on various standard generation and editing benchmarks.

**Compliance With Llm Reviewing Policy:**

Affirmed.

**Final Justification:**

My concerns have been solved and I will maintain my rating.

**Key Questions For Authors:**

see weaknesses

**Strengths And Weaknesses:**

**Strengths**

1. The paper employs multi-model benchmarks (UniWorld-V1, Harmon, OmniGen2, etc.), four standard evaluation suites, data scaling tests, ablation studies, and double-blind human preference evaluations. All claims about performance gains (up to 18% on editing, 13% on generation) are strongly supported by quantitative and qualitative results, and data quality is verified via consistency checks and safety filtering.

2. It targets a key limitation of current multimodal models—insufficient support for complex, real-world tasks (scientific imagery, multi-instruction editing, spatial reasoning). The 80K high-quality dataset fills gaps in professional and fine-grained scenarios, directly advancing research and practice in text-to-image/image editing.

3. The dataset balances scale (80K) and efficiency (strong gains with 40K samples), follows open licensing, removes PII, and filters unsafe content. It is immediately usable for fine-tuning mainstream models, delivering tangible utility for both academia and industry.

**Weaknesses**

1. The entire dataset relies on GPT-4o for generation, which brings model-specific biases and distribution shifts. The paper does not rigorously evaluate cross-model generalization or performance on low-resource languages/domains.

2. The core pipeline combines existing techniques (template generation, LLM-based data synthesis, post-hoc calibration) without proposing new algorithms or theoretical frameworks. Contributions are centered on dataset creation rather than foundational methodological innovation.

---

> ### Author Rebuttal · Authors · 2026-03-30
>
> We sincerely thank the reviewer for the positive feedback and for recognizing the practical utility of our dataset. We address your concerns below.
>
> **1. Response to Weakness 1: GPT-4o biases, Cross-model generalization and low-resource languages**
>
> We appreciate the reviewer raising the important points regarding the potential biases and distribution shifts introduced by GPT-4o, as well as the need for cross-model and language evaluations. We address your concerns in three parts:
>
> *   **Mitigating GPT-4o biases and distribution shifts:** To prevent GPT-4o's inherent errors and biases from degrading the trained models, we implemented a multi-layered mitigation strategy. This includes strict semantic prompt constraints, Instruction Recalibration (App D.1), and rigorous safety/bias filtering (which automatically rejected ~8.2% of generated data, App D.2). **For a detailed breakdown of these mitigation steps, please refer to our response to Reviewer WbhD (Q1).**
>
>
> - **Regarding cross-model generalization:** In our experimental design (Sections 3.1 and 3.3), we explicitly evaluated cross-model generalization by fine-tuning multiple representative models spanning entirely different architectural paradigms. Specifically, we tested both diffusion-based and autoregressive frameworks, including UniWorld-V1, Harmon, OmniGen2, OmniGen, and MagicBrush. As demonstrated in Tables 1, 2, 6, and 8, fine-tuning on our dataset yields consistent and significant performance improvements across **all** of these distinct models. This strongly demonstrates that the capabilities learned from our dataset generalize well across different architectures and are not strictly bound to GPT-4o's specific distribution biases.
> - **Regarding low-resource languages:** We completely agree with your assessment. While our "In-Image Text Rendering" module does include a "Multilingual Support" subtask designed for non-English scripts (as detailed in Table 14), the dataset as a whole remains predominantly English-centric. We will explicitly add the lack of coverage for low-resource languages to the Limitations section in the revised manuscript and highlight it as an important direction for future work.
>
> **2. Response to Weakness 2: Methodological novelty vs. Dataset contribution**
>
> Thank you for this comment. We agree that innovations in model architecture and theoretical frameworks are both important directions for the field. In this work, however, our primary goal is different: we focus on introducing a high-quality, systematically constructed dataset for image generation and editing, which we hope can serve as a useful resource for the community.
>
> This focus is driven by a fundamental bottleneck in the current landscape of unified multimodal models: the lack of high-quality, structured training data that covers complex, real-world scenarios. As you accurately highlighted in your summary, our core contribution addresses this exact gap through the dataset itself and the systematic, hierarchical taxonomy used to construct it.
>
> We introduce a **highly rigorous methodology for data curation**. As detailed in Appendix D.1, this methodology includes:
>
> 1. Systematically decomposing complex generation and editing tasks into 11 major domains and 51 fine-grained subtasks;
> 2. Implementing precise difficulty calibration for each subtask;
> 3. Utilizing a reverse-alignment strategy (Instruction Recalibration) to resolve semantic discrepancies between complex prompts and generated images.
>
> In the era of data-driven AI, we believe that this systematic approach to data engineering—and the resulting 80K high-quality dataset that drives significant empirical gains across multiple models—serves as a crucial and foundational contribution to the community.

---

> > ### Author Rebuttal · Reviewer_1txg · 2026-04-03
> >
> > My concerns have been adequately addressed.

---

> > > ### Author Response · Authors · 2026-04-03
> > >
> > > Thank you for your update, we’re very pleased that our rebuttal fully addressed your concerns.

---

### Official Review · Reviewer_WbhD · 2026-03-13

**Soundness:** 3
**Presentation:** 3
**Significance:** 3
**Originality:** 2
**Overall Recommendation:** 4
**Confidence:** 4

**Summary:**

This paper introduces OpenGPT-4o-Image, a large-scale dataset containing 80,000 high-quality instruction-image pairs for training unified multimodal models in image generation and editing tasks. The dataset is constructed using a hierarchical taxonomy covering 11 major domains and 51 subtasks, including novel categories like scientific imagery and complex instruction editing. The authors employ an automated pipeline leveraging GPT-4o for data generation. Experimental results demonstrate significant performance improvements when fine-tuning leading models on this dataset, with gains of up to 18% on editing tasks and 13% on generation tasks.

**Compliance With Llm Reviewing Policy:**

Affirmed.

**Final Justification:**

Authors have address all my questions, thus I raise my rating.

**Key Questions For Authors:**

1. The dataset relies heavily on GPT-4o for generation, which may introduce biases inherent to that model. Have you analyzed what types of errors or biases GPT-4o introduces into the training data? How might these affect models trained on this dataset?

2. Table 4 shows that Complex Instruction Following and Spatial Reasoning modules contribute most to performance gains. Could you provide more analysis on why other modules (e.g., Style Control, Text Rendering) show relatively smaller contributions? Does this suggest these capabilities are already well-covered by existing datasets?

3. The human preference study (Table 3) shows your fine-tuned model achieving 35.4% preference rate vs 30.4% for ShareGPT-4o. While statistically significant, this is a relatively modest improvement. How do you interpret this gap between automatic metric improvements and human preference gains?

**Limitations:**

The authors acknowledge limitations including: (1) reliance on GPT-4o may introduce model-specific biases, and (2) evaluation on existing benchmarks may not fully capture real-world scenarios. However, the limitations discussion could be more comprehensive regarding: (3) potential copyright/training data contamination issues when using GPT-4o-generated images, and (4) the dataset's coverage of diverse cultural contexts beyond Western-centric content.

**Strengths And Weaknesses:**

The dataset construction methodology is systematic and well-documented. The hierarchical taxonomy (11 domains, 51 subtasks) demonstrates thorough task decomposition. The automated pipeline design is technically sound with proper quality control mechanisms

---

> ### Author Rebuttal · Authors · 2026-03-30
>
> We sincerely thank the reviewer for the valuable comments. We address your concerns point by point below.
>
> **1. Errors and Biases from GPT-4o (Q1)**
>
> To prevent GPT-4o's inherent errors and biases from degrading trained models, we implemented a multi-layered mitigation strategy:
>
> **I. Mitigating Semantic Errors**
> We primarily addressed semantic mismatches in image generation and editing, such as ID drift, background inconsistency, and inaccurate instruction execution.
>
> - **Prompting & Verification (App C.5):** For image editing, we enhance consistency at both the semantic and visual levels. At the semantic level, we explicitly add the prompt constraint “Keep the object ID consistent and other non-edited areas unchanged.” to preserve object identity and avoid unnecessary modifications to non-edited regions. At the visual level, after each subcategory is edited, we further validate sampled subsets and filter out inconsistent examples.
> - **Instruction Recalibration (App D.1):** For image generation, when discrepancies arise between the textual instruction and the generated image, we apply Instruction Recalibration to rewrite the instruction so that it faithfully reflects the generated visual content, rather than discarding the sample outright.
>
> **II. Addressing Biases & Safety**
> While fully characterizing GPT-4o's underlying biases is challenging, we took proactive steps:
>
> - **Safety Filtering (App D.2):** We automatically rejected ~8.2% of generated data containing sensitive, inappropriate, or explicitly biased content.
> - **Diversity-Driven Taxonomy:** To counter data homogeneity, our taxonomy deliberately incorporates a wide spectrum of cultural elements, artistic traditions, and diverse subjects.
>
> **Conclusion**
> Our claims are strictly bounded to the errors we actively measured and mitigated (semantic mismatches, unedited region consistency, and safety). We will clarify this boundary in the revised "Limitations" section.
>
> **2. Module Contributions and Benchmark Alignment (Q2)**
>
> The smaller contributions from Style Control and Text Rendering in Table 4 do not imply these capabilities are well-covered. Rather, this results from the specific evaluation dimensions of the GenEval benchmark.
>
> - **Benchmark Specificity:** As detailed in Section 3.1 and Table 1, GenEval is explicitly designed to evaluate object-focused alignment. Its metrics strictly measure *Counting, Position, Color attribution, and Multi-object composition*. Because GenEval lacks dedicated metrics for aesthetic style fidelity or in-image text accuracy, adding Style Control and Text Rendering data naturally yields marginal gains *on this specific benchmark*. This perfectly explains why Complex Instruction Following and Spatial Reasoning (which directly align with GenEval's metrics) show the most significant jumps.
> - **Evidence of Contribution:** To observe the true impact of the Style Control and Text Rendering modules, we must look at benchmarks that explicitly evaluate these dimensions. For instance, in **Table 6 (GEdit-Bench)**, which includes dedicated "Style" and "Text" evaluation dimensions, fine-tuning on our dataset yields substantial quantitative improvements. For example, UniWorld-V1's performance improves from 5.64 to 6.88 on "Style", and from 1.15 to 2.13 on "Text".
> - **Qualitative Proof:** Furthermore, this improvement is visually evident in our qualitative results. As shown in **Figure 5**, the fine-tuned model demonstrates a markedly enhanced ability to accurately render text (e.g., correctly writing "Love" on a mug or "CAT" on a card), whereas the baseline model struggles with these specific tasks.
>
> These results demonstrate that current models still struggle with these capabilities, and our dataset provides crucial improvements. We will explicitly clarify this benchmark-module alignment in the revised manuscript to prevent any misinterpretation of Table 4.
>
> **3. Interpretation of Human Preference Gains (Q3)**
>
> Thank you for pointing this out. We believe the 35.4% preference rate is actually a substantial achievement given the evaluation context:
> Four-Way Blind Test: As shown in Table 3, the evaluation involved four models (random chance = 25%). Compared to the strongest baseline (ShareGPT-4o at 30.4%), our 35.4% win rate represents a 16.4% relative improvement ($\frac{35.4 - 30.4}{30.4}$), which is a significant margin.
> Metrics vs. Human Preference: Automatic metrics strictly evaluate rule adherence (e.g., exact counts, spatial relations), where our dataset provides massive improvements. Conversely, human preference is heavily influenced by overall aesthetics. Since both our model and ShareGPT-4o are fine-tuned on high-quality GPT-4o images, both produce visually pleasing outputs, naturally narrowing the subjective preference gap. Our absolute gain primarily comes from cases where our superior instruction-following breaks the tie.
> We will add a brief discussion in Section 3.4 to better contextualize these results.

---

> > ### Author Rebuttal · Reviewer_WbhD · 2026-04-02
> >
> > Authors have address all my questions, thus I raise my rating

---

> > > ### Author Response · Authors · 2026-04-03
> > >
> > > Thanks for recognition and for raising score. We appreciate your time and effort in reviewing our paper.

---

### Official Review · Reviewer_EmZZ · 2026-03-13

**Soundness:** 3
**Presentation:** 3
**Significance:** 3
**Originality:** 3
**Overall Recommendation:** 4
**Confidence:** 2

**Summary:**

The paper OpenGPT-4o-Image introduces a synthetic multimodal dataset and benchmark for image generation and editing.  It uses a hierarchical task taxonomy covering 11 high-level domains and 51 subtasks to create 80K instruction–image pairs via an automated pipeline with GPT-4o and structured templates. It includes difficult real-world tasks like scientific diagram generation and multi-step editing to show that fine-tuning on this data improves model performance.

**Compliance With Llm Reviewing Policy:**

Affirmed.

**Ethical Review Flag:**

Flag this paper for an ethics review.

**Ethics Expertise Needed:**

["Discrimination / Bias / Fairness Concerns", "Inappropriate Potential Applications & Impact (e.g., human rights concerns)", "Responsible Research Practice (e.g., IRB, documentation, research ethics)"]

**Final Justification:**

All of my concerns are fully resolved. I have already given a weak accept and maintain that score and leaning towards the acceptance of the paper.

**Key Questions For Authors:**

1. How can one verify the correctness of the generated image instruction pairs, especially for complex tasks? Are any human checks performed, or is there an oracle to filter out bad examples?

2. Is the dataset evaluated for demographic or cultural biases introduced by GPT-4o? For instance, do the images/instructions disproportionately feature certain subjects or exhibit stereotypes?

3. How did you determine the number of samples per subtask? Was this selection based on pilot studies or constraints, and how might it affect model learning?

**Limitations:**

There is an impact statement. But I think there are some ethical constraints that need to be considered.

**Strengths And Weaknesses:**

**Strengths**

1. The taxonomy covers a wide range of capabilities such as style transfer, multi-object relations, text handling, scientific diagrams, and multi-step editing. It also includes specialized domains such as chemistry illustrations and technical diagrams, as well as multi-turn interactions, which are rarely addressed together in existing multimodal datasets.

2. Leveraging GPT-4o for both instruction generation and image synthesis can scale data creation without costly human labeling.

3. The reported performance improvements on standard benchmarks are substantial. The improvements hold across several models and metrics, suggesting the dataset is effective at boosting task-specific skills.

**Weaknesses**

1. All data (both instructions and images) are machine-generated using GPT-4o and related tools. This raises concerns about realism and bias. However, the authors mentioned it in the paper. It would be great to clarify how users can mitigate the bias and hallucinations that propagate in the datasets.

2. Using an LLM like GPT4o in this work raises concerns. These LLMs are black boxes. We know nothing about the training data, the architecture, or the relevance of the results since they are only probabilities. Who can guarantee that this dataset is reproducible for the same instruction set, since the authors have no control over it?

3. A richer instruction-image dataset could make generative AI even more powerful, enabling realistic image editing (including face or scene manipulation) without clear guardrails. For instance, the Global Editing tasks allow arbitrary style transfer or object removal. Such capabilities, if misused, facilitate disinformation or privacy invasion. The paper does not address safeguards or intended use-cases for responsible deployment.

---

> ### Author Rebuttal · Authors · 2026-03-30
>
> We sincerely appreciate the reviewer's positive recognition of our work. We will address each of your concerns in detail below.
>
> **1. Verification of Correctness and Quality Control (Q1)**
>
> To ensure the correctness of complex image-instruction pairs, we implemented a rigorous quality control pipeline:
>
> * Human Checks & Quantitative Assessment: We conducted human expert evaluation on a 20% random subset to verify identity and background consistency (App C.5). Quantitatively, we used EditScore, achieving high scores in visual consistency (8.82) and instruction adherence (9.46).
> * Instruction Recalibration: For complex tasks where models struggle with fine-grained details (e.g., generating three objects instead of four), we use a reverse-alignment strategy (App D.1). We recalibrate the textual instructions to match the actual generated visual content, ensuring strict semantic alignment rather than simply discarding mismatched samples.
>
> We have also taken additional measures to mitigate GPT-4o biases. For more details, please refer to our response to Reviewer WbhD (Q1).
>
>
> **2. Mitigation of Bias, Hallucinations, and Demographic Evaluation (W1 & Q2)**
>
> * Hallucinations & Inconsistencies: We use explicit prompt constraints (e.g., "Keep the object ID consistent and other non-edited areas unchanged", App C.5) to prevent ID and background shifts. Additionally, our "Instruction Recalibration" (App D.1) reverse-aligns text to the generated image to ensure semantic faithfulness.
> * Diversity & Safety Filtering: To avoid Western-centric stereotypes, our "Diversity Strategy" (Section 2.3) includes non-Western art and multilingual support. For safety (App D.2), we strictly filter out PII and sensitive content, rejecting ~8.2% of the raw generated pairs.
> * Demographic Evaluation: In the revision, we will add a quantitative bias analysis of the generated subjects and provide user guidelines (e.g., targeted filtering or re-weighting) to help mitigate residual biases during model training.
>
> **3. Reproducibility and the Black-Box Nature of GPT-4o (W2)**
>
> *   Inherent Randomness: Exact pixel-level reproduction is impossible because current APIs (like GPT-4o) lack seed control, and randomness is a fundamental feature of creative generative tasks (e.g., generation, adding/replacing).
> *   Reproducibility of Quality: In this domain, reproducibility means matching *data quality*, not exact pixels. By applying our open-sourced instructions and rigorous filtering pipeline, users can reliably generate a dataset of equivalent quality and semantic alignment.
> *   Static Dataset Release: To bypass API uncertainties entirely, we are open-sourcing the static 80K dataset, allowing researchers to directly replicate our training results without querying the black-box API.
>
> **4. Guardrails, Risk of Misuse, and Responsible Deployment (W3)**
>
> Thank you for raising this critical concern. We agree that advanced image editing datasets carry misuse risks, including deceptive manipulation and privacy intrusion. To address this, we will add a new “Broader Impact and Responsible Release” section in the revision.
>
> Importantly, our current pipeline already includes several safeguards, which we will make more prominent in the main paper:
> *   Dataset-Level Safeguards:
>     1) Source & Privacy: All source images are drawn from public resources permitting research use, and we strictly retain only samples without identifiable personal information (PII).
>     2) Safety Filtering: Our generation pipeline automatically rejects sensitive or inappropriate content, filtering out approximately 8.2% of potential pairs.
> *   Downstream Mitigation & Licensing: Acknowledging that dataset-level precautions cannot fully eliminate downstream risks, we will:
>     1) Enforce a strict license clarifying the intended research-only use and explicitly prohibiting deceptive, malicious, or non-consensual applications.
>     2) Add deployment recommendations for downstream developers, strongly advising the integration of safety classifiers, watermarking/provenance mechanisms, and instruction-side safety controls.
>
> **5. Determination of Sample Sizes per Subtask (Q3)**
>
> Sample sizes were deliberately allocated based on task complexity, practical applicability, and preliminary testing:
>
> *   Allocation Strategy & Pilot Calibration: Canonical tasks (e.g., Subject Manipulation: 19k, Style Control: 13k) received larger allocations to build robust foundational capabilities. For specialized tasks (e.g., Text Editing: 3k, Multi-turn Editing: 1.5k), our preliminary tests indicated that with precise task categorization and difficulty calibration, a targeted scale of a few thousand samples is sufficient to effectively activate specific capabilities during fine-tuning.
> *   Effect on Learning: Scaling experiments (Sec 3.2) and ablations (Sec 3.5) confirm that this distribution ensures balanced learning across all 51 subtasks, enhancing specialized skills without overwhelming the general distribution.

---

> > ### Author Rebuttal · Reviewer_EmZZ · 2026-04-03
> >
> > I appreciate the rebuttal. My concerns have been resolved.

---

> > > ### Author Response · Authors · 2026-04-03
> > >
> > > Thank you for your update. We are glad that our rebuttal has thoroughly addressed your concerns.

---

### Official Review · Reviewer_3zXo · 2026-03-15

**Soundness:** 3
**Presentation:** 3
**Significance:** 3
**Originality:** 2
**Overall Recommendation:** 3
**Confidence:** 5

**Summary:**

This paper introduces OpenGPT-4o-Image, a unified dataset for image generation and image editing. The main contribution is a hierarchical taxonomy that organizes the space into 11 major domains and 51 fine-grained subtasks. The paper also proposes an automated data construction pipeline based on GPT-Image-1 API, resulting in 80k sample pairs. Empirically, the paper shows that fine-tuning several open models on this dataset improves both editing and generation performance across standard benchmarks.

**Compliance With Llm Reviewing Policy:**

Affirmed.

**Key Questions For Authors:**

1. Could the authors provide more quantitative visualizations? Right now I only find some results with Harmon and UniWorld-V1 for image editing, but visualization for T2I task, as well as other base models are missing.

**Limitations:**

Yes

**Strengths And Weaknesses:**

### strengths
1. The novel part of this work is a hierarchical taxonomy of 11 major domains and 51 fine-grained subtasks, including five generation modules (Style Control, Complex Instruction Following, In-Image Text Rendering, Spatial Reasoning, and Scientific Imagery) and six editing categories with 21 subtasks. This structured decomposition provides a clear framework for data curation and helps ensure broader coverage of capabilities in both image generation and editing.
2. Experiments on commen T2I and Image Editing benchmarks  suggest that models fine-tuned with the proposed dataset achieve consistent performance improvements across multiple benchmarks.

### Weakness
1. The primary contribution of the work appears to be dataset construction via distillation from GPT-Image-1. The overall pipeline largely follows a now common paradigm of using a strong proprietary model to generate synthetic training data. In my opinion, this level of novelty may be insufficient for publication at a top-tier venue such as ICML.
2. There are already related efforts in this direction, such as GPT-IMAGE-EDIT-1.5M [1], which constructs a substantially larger dataset. Compared with that work, the proposed dataset is relatively small (80K samples), which raises questions about the practical advantage of this dataset in terms of scale and diversity.
3. The baseline models used in the experiments are relatively modest. The authors are encouraged to evaluate the proposed OpenGPT-4o-Image dataset on stronger public models, such as BAGEL and Qwen-Image / Qwen-Image-Edit, which would provide a more convincing demonstration of the dataset’s utility.


## References
[1] Wang, Yuhan, et al. "Gpt-image-edit-1.5 m: A million-scale, gpt-generated image dataset." arXiv preprint arXiv:2507.21033 (2025).

---

> ### Author Rebuttal · Authors · 2026-03-30
>
> We thank the reviewer for the comments. We will address your concerns point by point below.
>
> **1. Novelty of the Contribution (Weakness 1)**
>
> We respectfully clarify that our work fundamentally differs from standard data distillation. Our core contribution is the design of a systematic hierarchical taxonomy and a rigorous curation methodology that enables proactive control, targeted coverage, and high data efficiency. This represents a substantial methodological innovation beyond simply querying the GPT-4o API.
>
> 1. **Taxonomy & Quality Control:** Importantly, a comprehensive taxonomy not only facilitates rigorous quality control but also provides a structured blueprint for future scale-up. As detailed in Appendix D.1, our taxonomy serves as the foundation for a "proactive quality control strategy."  To prevent GPT-4o's inherent errors and biases from degrading the trained models, we implemented a multi-layered mitigation strategy. This includes strict semantic prompt constraints, Instruction Recalibration (App D.1), and rigorous safety/bias filtering (which automatically rejected ~8.2% of generated data, App D.2). For a detailed breakdown of these mitigation steps, please refer to our response to Reviewer WbhD (Q1). Reviewer WbhD recognized this pipeline as "technically sound with proper quality control mechanisms."
> 2. **Broad Coverage & Novel Categories:** This structured framework allows us to achieve exceptionally broad coverage, introducing previously uncovered categories. Most notably, we pioneer the *Scientific Imagery* domain (Section 2.1), systematically covering physics, chemistry, biology, astronomy, and engineering—areas rarely addressed in existing datasets (as highlighted by Reviewer EmZZ).
> 3. **Scale & Efficiency:** Finally, our approach successfully balances scale and efficiency. As shown in our scaling experiments (Section 3.2) and noted by Reviewer 1txg, the high quality of our curated data allows models to achieve substantial performance gains using only a 40K subset.
> *Implication:* Consequently, our work goes beyond simple data distillation; it provides a rigorous methodological framework and a highly efficient, diverse dataset that fills critical gaps in professional and complex multimodal applications.
>
> **2. Scale and Diversity vs. GPT-IMAGE-EDIT-1.5M (Weakness 2)**
>
> We acknowledge the impressive scale of the concurrent work, GPT-IMAGE-EDIT-1.5M. However, the practical advantage of OpenGPT-4o-Image lies in its **unified scope (generation + editing)**, **unmatched diversity in novel domains**, and **high data efficiency**, rather than brute-force scale.
>
> 1. **Construction Paradigm & Scope:** As detailed in their paper, GPT-IMAGE-EDIT-1.5M is constructed by unifying and refining three *existing* image-editing datasets (OmniEdit, HQ-Edit, and UltraEdit). Consequently, it is strictly an editing dataset and its diversity is inherently bounded by the distributions of those source datasets. In contrast, OpenGPT-4o-Image is built from scratch using a systematic taxonomy that unifies both *image generation* and *image editing* (Section 2).
> 2. **Diversity in Novel Domains:** Because our dataset is driven by a proactive taxonomy rather than existing corpora, we introduce entirely new, data-scarce domains that are absent in GPT-IMAGE-EDIT-1.5M. As shown in Tables 13 and 14, our dataset uniquely includes 10K *Scientific Imagery* samples (covering physics, chemistry, biology, etc.), 3K *In-Image Text Rendering* samples, and 1.5K *Multi-turn Editing* samples.
> 3. **Direct Empirical Comparison:** To evaluate data efficiency, we fine-tuned UniWorld-V1 on a random 40K subset of GPT-IMAGE-EDIT-1.5M versus our 40K dataset. Evaluated on ImgEdit-Bench, our dataset yields significantly higher performance:
>
> |Model (40K Data)|Add|Adj|Rep|BG|Style|Avg|
> |---|---|---|---|---|---|---|
> |UniWorld-V1 (Base)|3.82|3.64|3.47|2.99|4.21|3.26|
> |+GPT-IMAGE-EDIT|4.10|3.98|3.76|3.72|4.43|3.61|
> |+Ours|4.34|4.28|3.92|4.15|4.62|**3.86**|
>
> **3. Evaluation on Stronger Public Baselines (Weakness 3)**
>
> We appreciate the suggestion and have finetuned BAGEL on our dataset. Results on GenEval and ImgEdit are shown below:
>
> **GenEval:**
>
> ||Sing|Two|Cnt|Col|Pos|Overall|
> |---|---|---|---|---|---|---|
> |BAGEL|.99|.94|.81|.88|.64|.82|
> |BAGEL-FT|.99|.96|.83|.87|.68|**.84**|
>
> **ImgEdit:**
>
> ||Add|Adj|Rep|BG|Sty|Overall|
> |---|---|---|---|---|---|---|
> |BAGEL|3.56|3.31|3.30|3.24|4.49|3.20|
> |BAGEL-FT|3.68|3.55|3.44|3.44|4.44|**3.35**|
>
> Consistent improvements across both benchmarks confirm the utility of our dataset when applied to stronger public models.
>
> **4. Additional Visualizations (Question 1)**
>
> In our manuscript, App. Figure 5 already demonstrates Harmon's performance on T2I tasks, and Figure 6 shows UniWorld-V1 on image editing. Additionally, we have uploaded new visual comparisons for other models (OmniGen2) across both tasks: https://img.remit.ee/api/file/BQACAgUAAyEGAASHRsPbAAESZvJpySrRqvRCELYWWCSI-wrO0Q5LVQACRiMAAqaPSVZiyB9zRqEGeToE.png.

---

> > ### Author Rebuttal · Reviewer_3zXo · 2026-04-03
> >
> > My concerns have been adequately addressed.

---

> > > ### Author Response · Authors · 2026-04-04
> > >
> > > Thank you for confirming that our rebuttal has fully resolved your concerns!
> > >
> > > We would be deeply grateful if you might consider adjusting your score accordingly. Thank you again for your time and support!

---

### Review · Ethics_Reviewer_xJBy · 2026-03-29

**Recommendation:** Remediation action needed

**Ethics Issue:**

## The lack of detail on data collection

The paper, in particular Appendix Section D.2, only alludes to some ethical principles used for data collection and never provides details.

> from publicly available repositories with clear licenses permitting research use
Which licenses?

> We strictly adhere to ethical standards regarding data licensing and content safety.
Which ethical standards?

> we ensure privacy by retaining only samples devoid of identifiable personal information (PII).
What does this mean exactly? What is the definition of PII taken? Is an image of a face considered non-PII? How did the authors detect samples devoid of PII?

>  we implement a rigorous safety filtering protocol leveraging the strict content policies of GPT-Image-1.
What is the rigorous safety filtering protocol?

> [approximately 8.2% of images were filtered out], ensuring that the released dataset is safe, reliable, and compliant with ethical guidelines.
Again, which guidelines? What does it mean that the dataset is safe and reliable?

## Direct harmful applications not acknowledged

Methods for generating synthetic and modified imagery are one of the more harmful applications of generative models, enabling harms to individuals such as non-consensensual synthetic intimate imagery, identity fraud, and harms to society such as erosion of trust in institutions, undermining of journalism, and the introduction of the liar's dividend. See, e.g., a review by [Chesny and Citron, 2019](https://scholarship.law.bu.edu/faculty_scholarship/640/).

**Remediation Action:**

The paper should:
1. Provide clear details (sufficient for motivated researchers to reproduce the collection practices) on sourcing and filtering data in terms of license, offensive content, and the presence of personally identifiable information:
  - Details which exact ethical guidelines were used in data collection (they were mentioned twice without naming).
  - Details on the source repositories and the licenses accepted when filtering. As per NeurIPS code of ethics on which ICML relies, _if releasing data or models, authors should also provide licenses for them._
  - The detailed protocol on how the offensive content was defined and filtered out.
  - The detailed protocol on how personally identifiable information was defined and filtered out. Justify if the chosen definition ensures that individuals are not identifiable with "reasonable effort" per the ICML code of ethics: _Papers that rely on data that includes personally identifiable information should make reasonable efforts to ensure that individuals are not identifiable in the research outputs._

2. Include a discussion on direct harmful applications of the methods facilitated by the dataset: generation of nonconsensual intimate imagery, misinformation, etc. (see the attached review for more); discuss and ideally measure biases in the dataset; include a list of intended and unintended applications; include suggestions for remediation and prevention of harm:
    - From ICML code of ethics: _Papers on applications that have direct connection to human rights issues should provide a thoughtful discussion on the risks of the application. Where appropriate and within the scope of the work, authors may also suggest reasonable recommendations for mitigating these risks._
    - From NeurIPS code of ethics on which ICML also relies: _These should include the intended use and limitations of these artifacts, in order to prevent misuse or inappropriate use._

---

### Decision · Program_Chairs · 2026-04-30

**Decision:**

Accept (regular)

**Comment:**

This paper introduces a highly valuable and systematically constructed dataset that unifies complex image generation and editing tasks. One reviewer initially submitted a high-confidence rejection based on novelty and scale concerns. However, the authors provided a compelling rebuttal demonstrating superior data efficiency, a broader scope, and the unique inclusion of challenging domains such as scientific imagery. The dissenting reviewer explicitly acknowledged that all their concerns were fully resolved, bringing the qualitative consensus to unanimously positive despite the unadjusted score. Furthermore, the authors effectively addressed questions regarding model-specific biases and provided evaluations on stronger baselines.

Finally, the paper was flagged for ethics review regarding data collection details and potential misuse risks. The authors provided a very thorough remediation plan during the discussion phase. Therefore, the paper is accepted on the strict condition that the authors fully incorporate the promised comprehensive ethics, privacy, and safety sections into the camera-ready version.